# Centriolar SAS-7 acts upstream of SPD-2 to regulate centriole assembly and pericentriolar material formation

Kenji Sugioka[1†], Danielle R Hamill[2†], Joshua B Lowry[1], Marie E McNeely[2], Molly Enrick[2], Alyssa C Richter[2], Lauren E Kiebler[2], James R Priess[3,4,5], Bruce Bowerman[1*]

[1]Institute of Molecular Biology, University of Oregon, Eugene, United States; [2]Department of Zoology, Ohio Wesleyan University, Delaware, United States; [3]Basic Sciences, Fred Hutchinson Cancer Research Center, Seattle, United States; [4]Molecular and Cellular Biology Program, University of Washington, Seattle, United States; [5]Department of Biology, University of Washington, Seattle, United States

**Abstract** The centriole/basal body is a eukaryotic organelle that plays essential roles in cell division and signaling. Among five known core centriole proteins, SPD-2/Cep192 is the first recruited to the site of daughter centriole formation and regulates the centriolar localization of the other components in *C. elegans* and in humans. However, the molecular basis for SPD-2 centriolar localization remains unknown. Here, we describe a new centriole component, the coiled-coil protein SAS-7, as a regulator of centriole duplication, assembly and elongation. Intriguingly, our genetic data suggest that SAS-7 is required for daughter centrioles to become competent for duplication, and for mother centrioles to maintain this competence. We also show that SAS-7 binds SPD-2 and regulates SPD-2 centriolar recruitment, while SAS-7 centriolar localization is SPD-2-independent. Furthermore, pericentriolar material (PCM) formation is abnormal in *sas-7* mutants, and the PCM-dependent induction of cell polarity that defines the anterior-posterior body axis frequently fails. We conclude that SAS-7 functions at the earliest step in centriole duplication yet identified and plays important roles in the orchestration of centriole and PCM assembly.

*For correspondence: bbowerman@molbio.uoregon.edu

†These authors contributed equally to this work

Competing interests: The authors declare that no competing interests exist.

## Introduction

The centrosome is an evolutionarily conserved non-membranous organelle that is important for microtubule organization in most animal cells. At its core is a pair of centrioles, each comprised of centriolar proteins and microtubules arranged in 9-fold radial symmetry around a structure called the cartwheel in most systems and what is thought to be its functional equivalent, the central tube, in *C. elegans* (*Boveri, 1900*; *Gönczy, 2012*). During mitosis, the two centrioles—termed the mother and daughter—are oriented orthogonally to each other and are surrounded by pericentriolar material (PCM), with the entire complex called the centrosome. The two centrosomes serve as microtubule organizing centers (MTOCs) that contribute to the assembly of the bipolar mitotic spindle during cell division (*Conduit et al., 2015*; *Woodruff et al., 2014*). In non-mitotic cells, centrosomes also contribute to cell motility, organelle positioning and intracellular transport (*Sugioka and Sawa, 2012*). The centriole also is called the basal body when it serves to nucleate and anchor the microtubule-based cilia that are present on most quiescent animal cells.

In proliferating cells, centrioles are duplicated in a cell cycle-dependent manner (*Robbins et al., 1968*; *Gönczy, 2012*), with two centrioles typically present at early interphase. Around the G1/S transition in many cells, or S phase in early *C. elegans* embryos, centriole duplication begins and a

procentriole forms orthogonally to each existing mother centriole. During S, G2 and the following cell cycle, the cartwheel becomes decorated with microtubules to form a 9-fold radially symmetric microtubule structure that elongates until it is equal in length to the mother centriole. As cells enter mitosis, PCM is recruited to the centriole, promoting microtubule nucleation and mitotic spindle assembly (*Woodruff et al., 2014*). Failure in the control of centriole duplication in humans is associated with cancer (*Zyss and Gergely, 2009*; *Gönczy, 2015*), microcephaly (*Bettencourt-Dias et al., 2011*; *Conduit et al., 2015*), and ciliopathies such as Bardet-Biedl and Oral-Facial-Digital syndromes (*Forsythe and Beales, 2013*; *Thauvin-Robinet et al., 2014*).

Although the centriole duplication cycle has been known for decades, the molecular mechanisms that govern this fundamental process have only recently begun to be understood (*Dong, 2015*; *Fu et al., 2015*). Much of the foundational work has come from molecular genetic studies in *C. elegans* that have identified a requirement for the sequential recruitment of five proteins—one kinase and four coiled-coil proteins—called core components (*Delattre et al., 2006*; *Pelletier et al., 2006*) The earliest acting of these core components is the coiled-coil protein SPD-2 (*Delattre et al., 2006*; *Kemp et al., 2004*; *Pelletier et al., 2004*; *2006*), which is required for the recruitment of ZYG-1, a polo-like kinase that phosphorylates and recruits SAS-6 to the daughter centriole (*Kitagawa et al., 2009*; *O'Connell et al., 2001*). SAS-6, together with SAS-5, promotes formation of the daughter centriole's central tube, which is similar to the cartwheel in other animals (*Dammermann et al., 2004*; *Delattre et al., 2004*; *Leidel et al., 2005*; *Pelletier et al., 2006*). SAS-6 and SAS-5 also recruit SAS-4, which plays a role in centriolar microtubule assembly around the central tube (*Kirkham et al., 2003*; *Leidel and Gönczy, 2003*; *Pelletier et al., 2006*). These core proteins are conserved by sequence or functional homology to counterparts in other organisms (*Carvalho-Santos et al., 2010*; *Hodges et al., 2010*). For example, SPD-2 is homologous to Cep192 in mammals and is required for the centriolar localization of other core components in both *C. elegans* (*Delattre et al., 2006*; *Pelletier et al., 2006*) and humans (*Kim et al., 2013*; *Zhu et al., 2008*). However, the mechanism that recruits SPD-2 to initiate centriole duplication remains unknown.

We isolated *or452*ts, a conditional reduction-of-function mutant that fails to assemble bipolar mitotic spindles in early *C. elegans* embryos. We identified *sas-7*/T07C4.10, which encodes a predicted coiled-coil protein, as the gene mutated in *or452*ts mutants. Using transmission electron microscopy (TEM), we found that SAS-7 is required for centriole duplication and elongation, for the formation of a peripheral centriole structure we call the paddlewheel, and for PCM assembly. Interestingly, our genetic analysis suggests that SAS-7 is required for daughter centrioles to become duplication competent and for mother centrioles to maintain competence. We also have found that SAS-7 protein can bind SPD-2, and is required for SPD-2 centriolar localization. However, SAS-7 localizes to centrioles independently of SPD-2, in contrast to the other four core centriolar components. Thus SAS-7 acts upstream of SPD-2 in centriole duplication, and may recruit SPD-2 to the centriole.

## Results

### A mutation in the *sas-7* gene causes bipolar spindle assembly defects

We isolated the mutant allele *or452*ts after chemical mutagenesis in a genetic screen for temperature sensitive embryonic-lethal mutations that result in early embryonic cell division defects (*Encalada et al., 2000*). At the restrictive temperature, mutant cells entered mitosis normally. However, the condensed chromosomes were often distributed around the periphery of a single microtubule aster (*Figure 1A and B*, *Table 1* and *Figure 1—figure supplement 1A*), rather than being captured midway between the two poles of the bipolar mitotic spindles that assemble in wild-type embryos. We also used immunofluorescence to detect the PCM component SPD-5 (*Hamill et al., 2002*) and the centriole component SAS-4 (*Kirkham et al., 2003*; *Leidel and Gönczy, 2003*) and similarly observed monopolar spindles during the first mitotic division of most embryos, in the cell called $P_0$ (*Figure 1A and B* and *Figure 1—figure supplement 1A*). While 12% of the mutant embryos proceeded apparently normally to the 4 cell stage, they invariably failed to assemble bipolar spindles in some subsequent divisions, and *or452*ts embryos arrested as mostly undifferentiated balls of variably sized cells (*Table 1*). As the *or452*ts mutant phenotype is similar to other spindle assembly abnormal (*sas*) mutants, we named the affected gene *sas-7*.

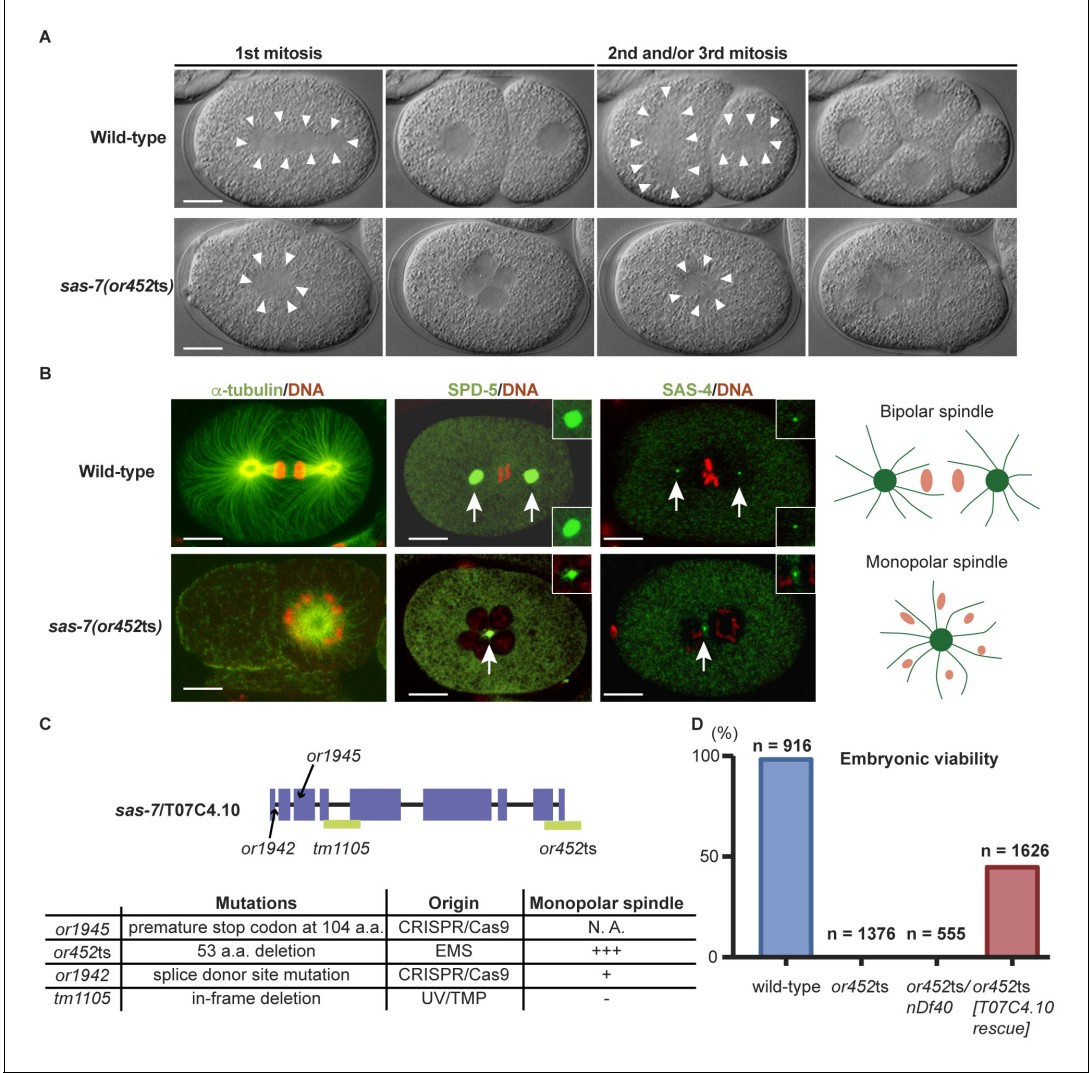

**Figure 1.** SAS-7 is required for bipolar mitotic spindle assembly. (**A**) Time-lapse DIC images of the first, second and third mitotic cell divisions in wild-type and *sas-7(or452*ts) mutants. Arrowheads indicate the shape of the mitotic spindle. (**B**) Immunofluorescence of mitotic spindle components. α-tubulin, SPD-5 and SAS-4 were stained by antibodies, shown in green and indicated by arrows. Insets are magnified views. DNA staining is red. (**C**) *sas-7* gene model and mutant alleles. See Materials and Methods for details. (**D**) Embryonic viability for indicated genotypes. Scale bars indicate 10 μm.

The following figure supplements are available for figure 1:

**Figure supplement 1.** SAS-7 is required for bipolar spindle formation.

**Figure supplement 2.** RNAi knock-down against SAS-7 is not effective.

**Figure supplement 3.** SAS-7 N-terminus has weak homology to human SPERT protein.

**Figure supplement 4.** SAS-7 C-terminus has weak homology to human CFAP57/WDR65 protein.

To identify the physical location of *sas-7* in the genome, we used a combination of genetic mapping, whole genome sequencing and complementation. Three-factor mapping positioned the allele at 2.4 ± 0.5 cM on the right arm of chromosome III, but there were no obvious candidate genes in this region, based on phenotypic analysis of existing mutants, DNA sequencing, genetic complementation tests, predicted gene functions, and reported RNAi phenotypes. We then used simultaneous whole-genome sequencing and SNP mapping (*Doitsidou et al., 2010*; *Minevich et al., 2012*) to

**Table 1.** Embryonic viability and early embryonic phenotypes.

| | Embryonic Viability (%)[*] | | Monopolar spindle[†] | | | | | | | Multi-nucleated? |
|---|---|---|---|---|---|---|---|---|---|---|
| | 15°C | 26°C | P0 1 st mitosis | AB 2nd mitosis | P1 3rd mitosis | ABa 4th mitosis | ABp 5th mitosis | EMS 6th mitosis | P2 7th mitosis | Yes/No |
| Wild-type | 97.8 (n = 9, 2010) | 98.3 (n = 9, 916) | 0% | 0% | 0% | 0% | 0% | 0% | 0% | No |
| sas-7 (or452ts) | 13.1 (n = 10, 1838) | 0 (n = 11, 1376) | 62% (24/39) | 47% (7/15) | 40% (6/15) | 20% (1/5) | 80% (4/5) | 80% (4/5) | 60% (3/5) | Yes |
| sas-7 (or1942) | 0 (n = 7, 305) | | 0% (0/15) | 4% (1/26) | 0% (0/26) | 29% (6/21) | 24% (5/21) | 33% (7/21) | 14% (3/21) | Yes |
| sas-7 (tm1105) | 96.9 (n = 7, 632) | 20 (n = 15, 533) | 0% (0/15) | 0% (0/15) | 0% (0/15) | 0% (0/12) | 0% (0/12) | 0% (0/12) | 0% (0/12) | Yes |
| sas-7 (or452ts) /nDf40[‡] | N. D. | 0 (n = 15, 555) | 56% (19/34) | 50% (12/24) | 29% (7/24) | N. D. | N. D. | N. D. | N. D. | Yes |
| sas-7 (or452ts) /+ | N. D. | 97.2 (n = 9, 1173) | 0% (0/11) | 0% (0/11) | 0% (0/11) | N. D. | N. D. | N. D. | N. D. | No |
| or452ts female /wild-type male[§] | N. D. | 7.31 ± 2.28 (n = 7, 711) | 0% (0/20) | 0% (0/31) | 0% (0/31) | 24% (10/41) | 44% (18/41) | 42% (17/41) | 17% (7/41) | Yes |
| Wild-type female /or452ts male[#] | N. D. | 57.4 ± 3.54 (n = 19, 1669) | 65% (13/20) | 0% (0/13) | 0% (0/12) | 0% (0/12) | 0% (0/12) | 0% (0/12) | 0% (0/12) | Yes |
| Wild type female /wild-type male[¶] | N. D. | 97.1 ± 1.09 (n = 6, 938) | N. D. | N. D. | N. D. | N. D. | N. D. | N. D. | N. D. | N. D. |

[*]L1 stage larvae were grown at the permissive temperature (15°C) or restrictive temperature (26°C). Once gravid, single worms were transferred every 12–24 hr for ~3 days, and the embryos laid and hatched were scored. Percent hatching is given ± standard error of the mean. In parenthesis are the number of worms scored and the number of embryos counted.

[†]L1 stage larva were grown at 26°C until young adults and videos were made typically starting prior to the pronuclei meeting.

[‡]nDf40 deletion, chr III: 0.77–3.36.

[§]Female genotype: sas-7(or452ts);fem-1(hc17ts).

[#]Female genotype: fem-1(hc17ts), Male genotype: sas-7(or452ts); him-8(e1489). Worms were raised at 26°C from L1 stage.

[¶]Female genotype: fem-1(hc17ts), Male genotype: him-8(e1489). Worms were raised at 26C from L1 stage.

identify T07C4.10 as a candidate gene, and found that a wild-type T07C4.10 DNA construct injected into sas-7(or452ts) mutant worms partially rescued embryonic viability (**Figure 1D**), confirming gene identity. T07C4.10 is predicted to have nine exons and to encode a 1014 amino acid, coiled-coil protein that lacks clear homology to proteins in other animal phyla (**Figure 1C**; see Discussion). The sas-7(or452ts) allele has a 456 base pair deletion beginning in the middle of exon 8, predicted to remove the last 53 amino acids of the protein and nucleotides downstream of the coding region (**Figure 1C**).

We made additional mutant alleles of sas-7/T07C4.10 using CRISPR/Cas9 genome editing (Materials and methods), because RNAi against sas-7 was not effective (**Figure 1—figure supplement 2**), and an available allele of T07C4.10, tm1105, has an in-frame deletion of 64 codons and was homozygous viable with no apparent early embryonic defects (**Figure 1—figure supplement 2** and **Table 1**). We obtained two new alleles, or1945 and or1942, and confirmed that sas-7/T07C4.10 is essential and required for bipolar spindle formation. The allele or1945 has a premature stop at codon 104 and resulted in adult sterility (**Figure 1C**). The or1942 allele has a point mutation in the splice donor site of intron one and resulted in embryonic lethality with a weakly penetrant monopolar spindle phenotype in early embryos (**Figure 1C** and **Figure 1—figure supplement 1B**). We also found that worms with one copy of sas-7(or452ts), and one copy of a chromosomal deficiency (nDf40) that completely deletes the T07C4.10 gene, produced embryos with a monopolar spindle phenotype similar in penetrance to that observed in embryos from sas-7(or452ts) homozygotes (**Figure 1D** and

*Table 1*). We conclude that *sas-7(or452*ts) is a recessive, reduction of function allele of the T07C4.10 gene and used this mutant allele for the rest of the experiments reported here.

## SAS-7 is required for centriole duplication in both meiosis and mitosis

The oocytes in self-fertile *C. elegans* hermaphrodites lose their centrioles during oogenesis (*Mikeladze-Dvali et al., 2012*), and upon fertilization the sperm contributes a pair of centrioles that duplicate within the egg cytoplasm by recruiting maternally supplied proteins (*Delattre and Gönczy, 2004*). Therefore, the monopolar spindle phenotypes observed in *sas-7(or452*ts) mutants during the first embryonic mitosis in the $P_0$ cell, and during the second and third mitoses in the AB and $P_1$ cells, suggest that there are centriole duplication defects during both male meiosis and the first embryonic mitosis, respectively (*Figure 2A*). To count the number of centrioles, we stained embryos with an antibody that recognizes SAS-4. As centrioles are initially unseparated after duplication, we focused on $P_0$ cells at prophase to metaphase and AB/$P_1$ cells at interphase to anaphase to count the number of separated centrioles after male meiosis II and after the first embryonic mitosis, respectively. We found that in *sas-7(or452*ts) mutants, SAS-4 formed only one focus in both $P_0$ and in AB and $P_1$, suggesting that centriole duplication during male meiosis II and the first embryonic mitosis were both defective (*Figure 2B and C*). To further test this hypothesis, we used genetic crosses to look for possible oocyte and sperm SAS-7 requirements (*Figure 2D* and *Table 1*). When we crossed homozygous *sas-7(or452*ts) males with feminized (*fem-1*) hermaphrodites lacking sperm, the $P_0$ mitotic spindles were monopolar in 65% of the embryos but later divisions were normal (*Figure 2D* and *Table 1*). On the other hand, when we mated *sas-7*(or452ts);*fem-1* females with wild-type males, the $P_0$, AB and $P_1$ spindles were always bipolar, but the mitotic spindles in subsequent divisions were often monopolar (*Figure 2D* and *Table 1*). These results suggest that paternal and maternal SAS-7 contribute to centriole duplication during male meiosis and embryonic mitosis, respectively.

While our results indicate that sperm meiosis II is defective, we did not detect any centriolar duplication defects during sperm meiosis I in *sas-7(or452*ts) mutants (*Figure 2—figure supplement 1*). Similarly, defects in sperm meiosis I have not been reported in other centriole duplication-defective mutants. However, because the *or452*ts allele causes only a partial loss of gene function, we cannot conclude that SAS-7 is not required for sperm meiosis I: putative null *sas-7(or1945)* mutants are adult sterile, as are other strong centriole duplication-defective mutants (*Mikeladze-Dvali et al., 2012*). Thus it is possible that centriole duplication during mitosis in germline stem cells and during sperm meiosis I require lower levels of these gene functions compared to sperm meiosis II and mitosis.

Taken together, our results suggest that reduced SAS-7 activity resulted in severe centriole duplication defects in both sperm meiosis II and embryonic mitosis.

## SAS-7 is required for a daughter to mother centriole transition in duplication competence and for the maintenance of duplication competence

When we mated *sas-7(or452*ts);*fem-1* females with wild-type males, not only the first ($P_0$) division but also the second (AB) and third ($P_1$) divisions were normal, unlike in other centriole duplication mutants (*O'Connell et al., 2001*; *Delattre et al., 2004*), indicating that wild-type sperm centrioles retain the ability to duplicate in the absence of maternal SAS-7. To further investigate this observation, we focused on centriole duplication in the subsequent embryonic cell cycles: the fourth (ABa and ABp), sixth (EMS) and seventh ($P_2$) cell divisions. We considered two models for centriole duplication. In the first model, the centrioles in 2 cell stage sperm(+), oocyte(-) *sas-7* mutant embryos randomly fail to duplicate with a probability of p=a (random defects model; *Figure 3A*). In the second model, mother centrioles in the sperm(+), oocyte(-) mutant embryos retain the ability to duplicate while daughter centrioles become incompetent with a probability of p=b (daughter-mother transition defects model; *Figure 3B*). In the latter model, no 4 cell stage embryos with three or four monopolar spindles would be expected. Furthermore, these two models predict very different distributions of centriole duplication defects in 4 cell stage sperm(+), oocyte(-) embryos that have two cells with bipolar and two with monopolar spindles. In the random defects model (*Figure 3A*), 34% of the embryos would have duplication defects in both anterior (ABa and ABp) or both posterior ($P_2$ and EMS) cells; note that this pattern does not depend on the value of a. However, in the daughter-

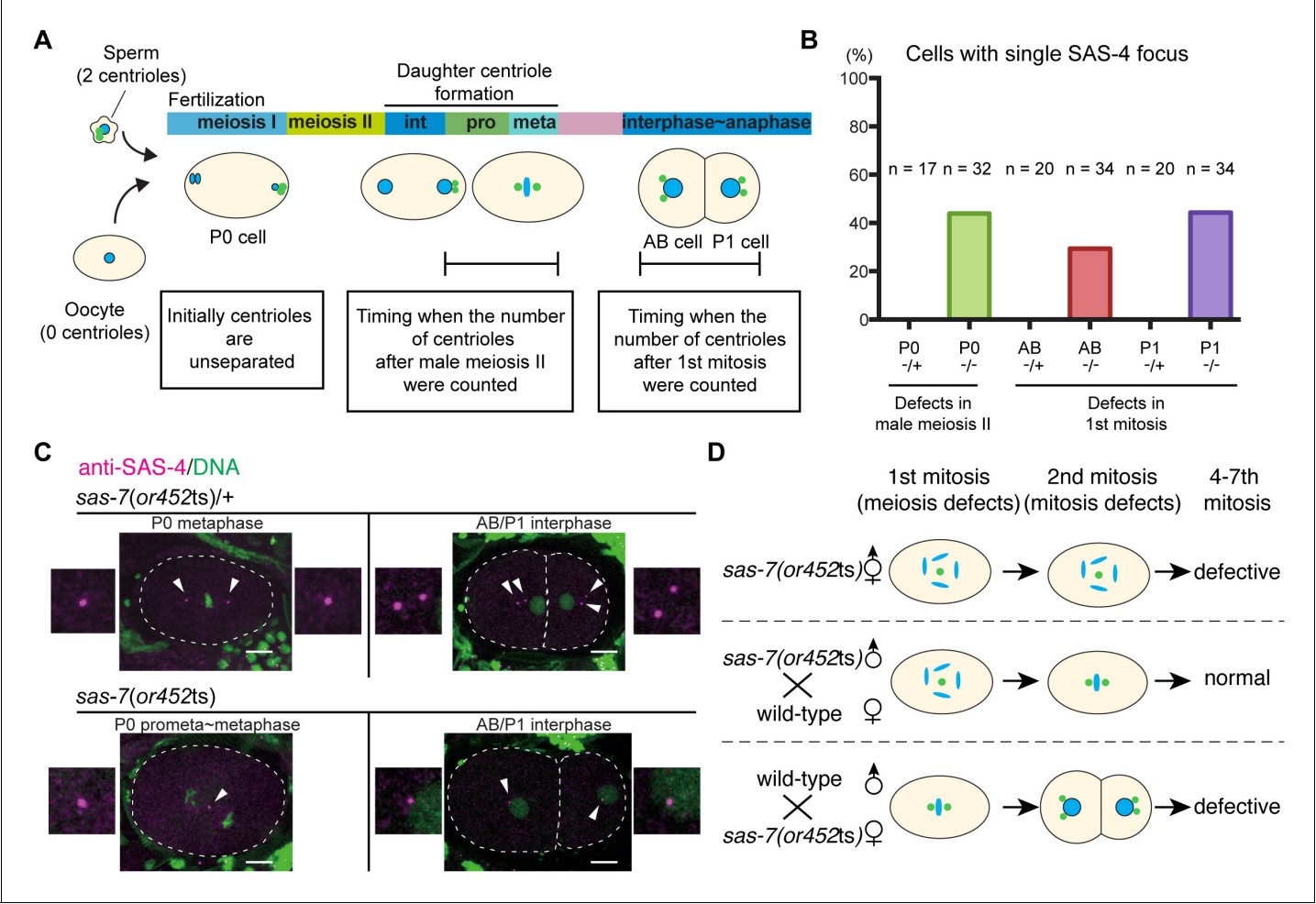

**Figure 2.** SAS-7 regulates centriole duplication during both sperm meiosis and embryonic mitosis. (**A**) Schematic illustration of the timing of centriole duplication and times at which centriole numbers were counted. As duplicated centrioles are initially unseparated, the number of centrioles after male meiosis II and the first mitosis were monitored after centriole separation at the time intervals indicated in the text boxes. (**B**) Percentages of cells with single SAS-4 foci; embryos were self-progeny either from control heterozygous (-/+) hermaphrodites, or from homozygous (-/-) mutant hermaphrodites, as indicated below cell names on x-axis. Prophase to metaphase $P_0$, and interphase to anaphase AB and $P_1$ cells, were scored for SAS-4 foci numbers. (**C**) SAS-4 foci number after male meiosis II and the first embryonic mitosis. SAS-4 and DNA are magenta and green, respectively. Arrowheads and dotted lines indicate centrioles and cell outlines, respectively. Scale bars indicate 10 μm. (**D**) Paternal and maternal contributions of *sas-7* gene activities to centriole duplication. Paternal and maternal SAS-7contribute to centriole duplication in male meiosis II and mitosis, respectively. See also *Table 1*.

The following figure supplement is available for figure 2:

**Figure supplement 1.** SAS-7 localizes to sperm throughout spermatogenesis but *sas-7(or452*ts) mutants did not show meiosis I defects.

mother transition defects model (*Figure 3B*), all such embryos would have monopolar spindles in one anterior and one posterior cell, regardless of the value of b. The distributions of monopolar spindles in sperm(+), oocyte(-) *sas-7(or452*ts) 4 cell stage embryos with two monopolar spindles strongly support the daughter-mother transition defects model (*Figure 3C*). We therefore conclude that SAS-7 is required for daughter centrioles to acquire duplication competence and become mother centrioles.

Our genetic data also suggest that SAS-7 is required for mother centrioles to maintain duplication competence. As described above, in sperm(-), oocyte(-) *sas-7(or452*ts) mutants, spindle formation during sperm meiosis I was normal, suggesting that the mother centrioles were duplication competent. In meiosis II, centriole duplication sometimes failed and resulted in the formation of monopolar spindles in the $P_0$ cell. As one primary spermatocyte produces four mature sperm, if only daughter

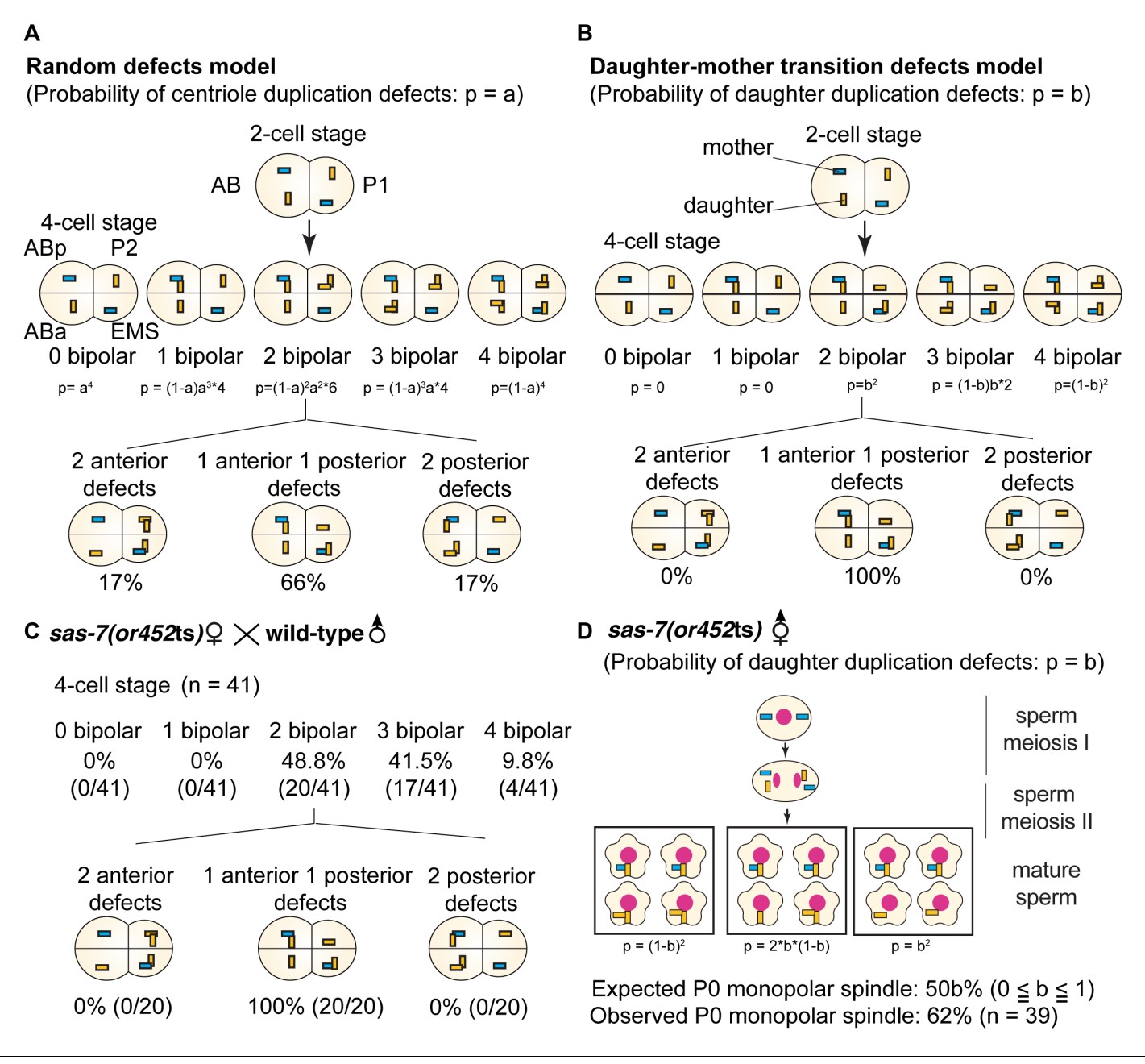

**Figure 3.** SAS-7 is required for newly formed daughter centrioles to become mother centrioles. (A) Random defects model for centriole duplication in *sas-7* mutants. If both mother and daughter centrioles fail to duplicate with a probability of a (p=a), the distribution of 4 cell stage monopolar spindles will be as shown. (B) Daughter-mother transition defects model for *sas-7* mutants. If mother centrioles are never defective for centriole duplication while daughter centrioles fail to duplicate with a probability of b (p=b), the distribution of 4 cell stage monopolar spindles will be as shown. (C) Distribution of 4 cell stage monopolar spindles in sperm(+), oocyte(-) embryos from *sas-7(or452*ts) mutant females crossed with wild-type males. (D) SAS-7 may also be required for mother centrioles to maintain duplication competence. In *sas-7(or452*ts) mutants, centriole duplication in meiosis I is normal but is sometimes defective in meiosis II resulting in the $P_0$ monopolar spindle phenotype. However, the observed $P_0$ monopolar spindle penetrance (62%; see *Table 1*) is higher than even the case where all daughter centrioles fail to duplicate (b = 1.0, in which case all sperm would have centrioles depicted in the right-most box, such that 50% of $P_0$ spindles would be monopolar); even fewer would be monopolar at lower values of b. These results suggest that SAS-7 is required for some of the mother centrioles to duplicate in sperm(-), oocyte(-) *sas-7(or452*ts) embryos. Mother centrioles are blue; daughters are orange.

The following figure supplement is available for figure 3:

**Figure supplement 1.** *sas-7(or452*ts) mother centrioles lose duplication competence.

centrioles lost duplication competence with a probability of b, 50b% of $P_0$ cells would be expected to have a monopolar spindle. However, the observed penetrance of the $P_0$ monopolar spindle phenotype was 62% (*Table 1*), suggesting that in addition to daughter centrioles, mother centrioles also lost duplication competence (*Figure 3D*). The same conclusion follows from a consideration of the AB and $P_1$ cell divisions. If the daughter-to-mother transition model is applicable, at most 11% of the embryos would be expected to have monopolar spindles in both AB and $P_1$ (*Figure 3—figure supplement 1*). But we saw monopolar spindles in both AB and $P_1$ in 27% of the sperm(-), oocyte(-) *sas-7(or452*ts) embryos (*Table 1*), again suggesting that in addition to daughter-to-mother transition defects, SAS-7 is required for mother centrioles to maintain their acquired duplication competence.

## SAS-7 is required for centriole assembly and elongation

Although immunofluorescence detected only a single focus of centriolar SAS-4 protein in fixed *sas-7 (or452*ts) embryos (*Figure 1B*), we cannot distinguish centriole duplication and separation defects using conventional light microscopy, due to the small size of the centriole (~100 nm in diameter). We therefore used transmission electron microscopy (TEM) to count and analyze centrioles in wild-type and mutant embryos during the first three mitotic cycles. We found that 90% of the wild-type centrosomes contained a pair of centrioles (n = 20); 10% contained only a single centriole and appeared to be at early cell cycle stages. In contrast, only 35% of the *sas-7(or452*ts) centrosomes contained a pair of centrioles, whereas the remaining 65% contained a single one (n = 17) (*Figure 4A*), similar to the penetrance of monopolar spindles observed by DIC microscopy (*Table 1*). We conclude that *sas-7* is required for centriole duplication.

Consistent with previous ultrastructural studies of *C. elegans* centrioles, the wild-type centrioles had nine singlet microtubules surrounding a central tube about 60 nm diameter. This 'central tube' is thought to be the functional equivalent of the cartwheel described in other eukaryotes, although there appear to be some structural differences (*Figure 4B and C* and *Figure 4—figure supplement 1*) (*Gönczy, 2012*; *Pelletier et al., 2006*). In other eukaryotes, the essential cartwheel protein SAS-6 oligomerizes to form stacked rings that serve as a template for procentriole formation (*Kitagawa et al., 2011*; *van Breugel et al., 2011*). While SAS-6 is conserved in *C. elegans*, previous studies have not reported the existence of a *C. elegans* cartwheel. Here we found that 61% of the wild-type centrioles contained a distinct inner tube of about 18.3 ± 2.9 nm (n = 11) in diameter, with an electron-lucent center (*Figure 4B and C* and *Figure 4—figure supplement 1*), and in some images radial spokes appeared to extend outward from the inner tube (*Figure 4B*). This inner tube has not been described in *C. elegans*, but it appears to be present in TEM images from a previous study although its presence was not discussed (*O'Connell et al., 2001*). Based on its position, size and morphology, the inner tube resembles the central hub of the cartwheel described in other eukaryotic centrioles, which is thought to be comprised of an aligned stack of SAS-6 cartwheels. The *C. elegans* inner tube we describe is similar in diameter to the central hubs in other organisms: 25 nm diameter in *Chlamydomonas reinhardtii* (*Cavalier-Smith, 1974*; *Geimer and Melkonian, 2004*), 22 nm in *Trichonympha* (*Guichard et al., 2012*), about 16 nm in fly spermatocytes (*Roque et al., 2012*) and 20 nm in human lymphocytes (*Guichard et al., 2010*). Thus, we suggest that the 18 nm inner tube of the *C. elegans* centriole is homologous to the central hub in other centrioles.

We also detected electron-dense protrusions that extend from each centriolar microtubule along the entire length of the centriole (*Figure 4B*). Protrusions that might correspond to these structures have been observed previously in *C. elegans* and termed 'appendages', but they were not extensively characterized (*Pelletier et al., 2006*). We propose to rename this structure a paddlewheel based on its appearance, to avoid confusion with the mammalian appendages that extend only from the distal tip of mother centrioles and play important roles in cilia formation (*Tanos et al., 2013*) but are dispensable for cell division (*Ishikawa et al., 2005*). We found that each protrusion of the paddlewheel extended 30.9 ± 4.62 nm from the center (n = 17) and was twisted in a clockwise direction when viewed from the distal end (*Figure 4B and C*), which is the same chirality observed for centriolar microtubule doublets and triplets in other organisms (*Geimer and Melkonian, 2004*; *Gönczy, 2012*; *González et al., 1998*; *Paintrand et al., 1992*).

We next analyzed in detail the ultrastructure of *sas-7(or452*ts) centrioles. Interestingly, many of the mutant centrioles had very reduced paddlewheels, resulting in a smaller centriole diameter (*Figure 4B and D* and *Figure 4—figure supplement 1B*). Furthermore, *sas-7* mutant centrioles were

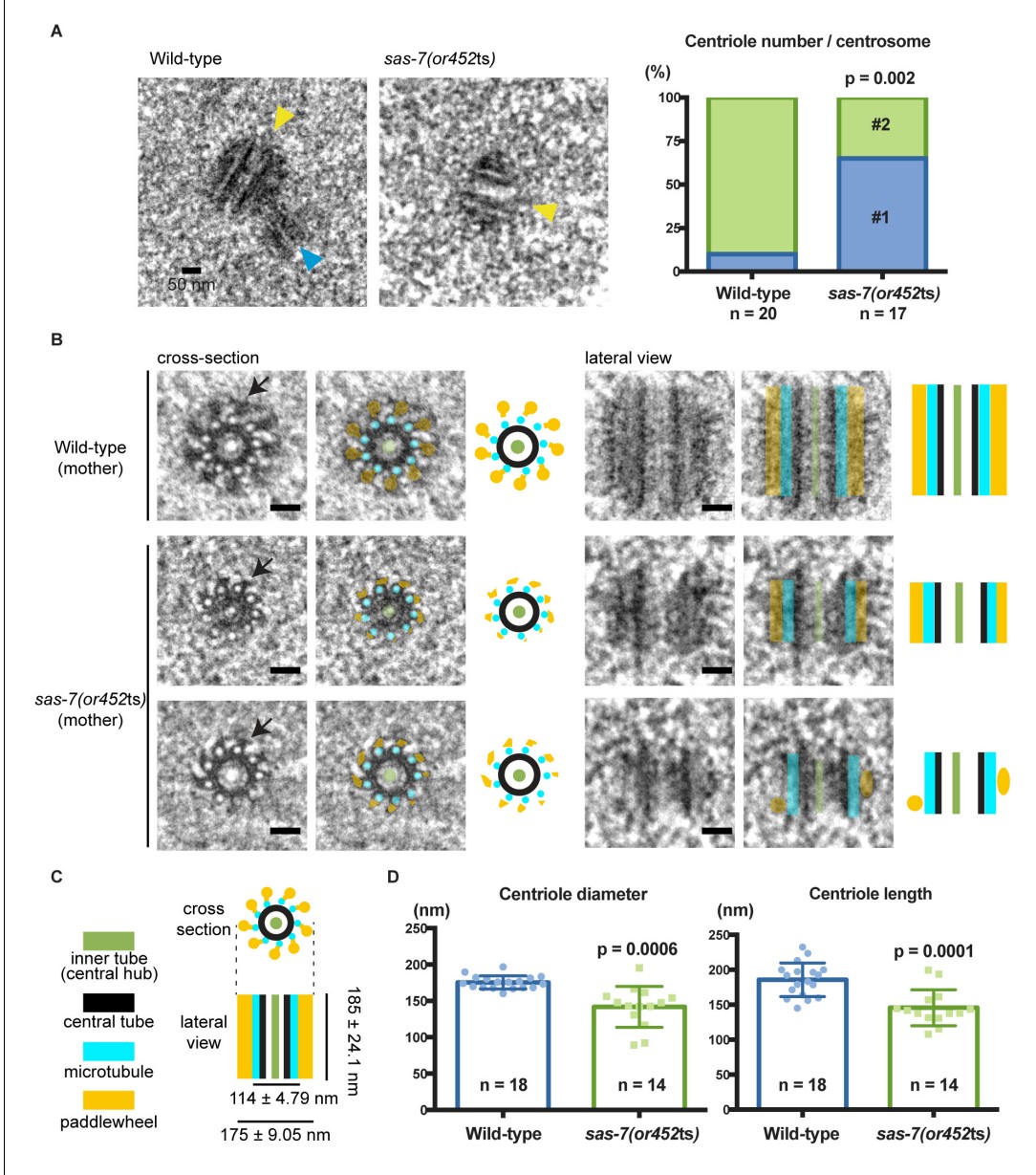

**Figure 4.** SAS-7 is required for centriole duplication, assembly and elongation. (**A**) TEM images of centrioles. Yellow and blue arrowheads indicate the mother and daughter centrioles, respectively. P-value in the graph was calculated by the Fisher's exact test. (**B**) Cross-section and lateral view TEM images of centrioles. Arrows indicate the paddlewheel structure. Colored overlays indicate the distinct structural units shown in C. (**C**) Schematic representations of *C. elegans* centriole structure and chirality when viewed from the distal end. (**D**) Quantified centriole diameter and length. Mean ± SD shown. P-values calculated by the Welch's t-test. See also *Figure 4—figure supplement 1*. Scale bars indicate 50 nm.

The following figure supplement is available for figure 4:

**Figure supplement 1.** Quantification of centriole structural units.

shorter than those in wild-type embryos (*Figure 4B and D*). We conclude that SAS-7 is required not only for centriole duplication, but also for paddlewheel assembly and centriole elongation.

## SAS-7 is a new centriolar component that does not require SPD-2 for its localization

To analyze the sub-cellular localization of endogenous SAS-7 protein, we inserted GFP coding sequences into the *sas-7* genomic locus using CRISPR/Cas9 mediated homologous recombination (Materials and methods). GFP::SAS-7 homozygous strains were viable (94.9% embryonic viability, n = 1249), suggesting that the GFP fusion protein is functional. We detected GFP::SAS-7 at centrosomes throughout spermatogenesis, embryogenesis, and in the germline but not in mature oocytes

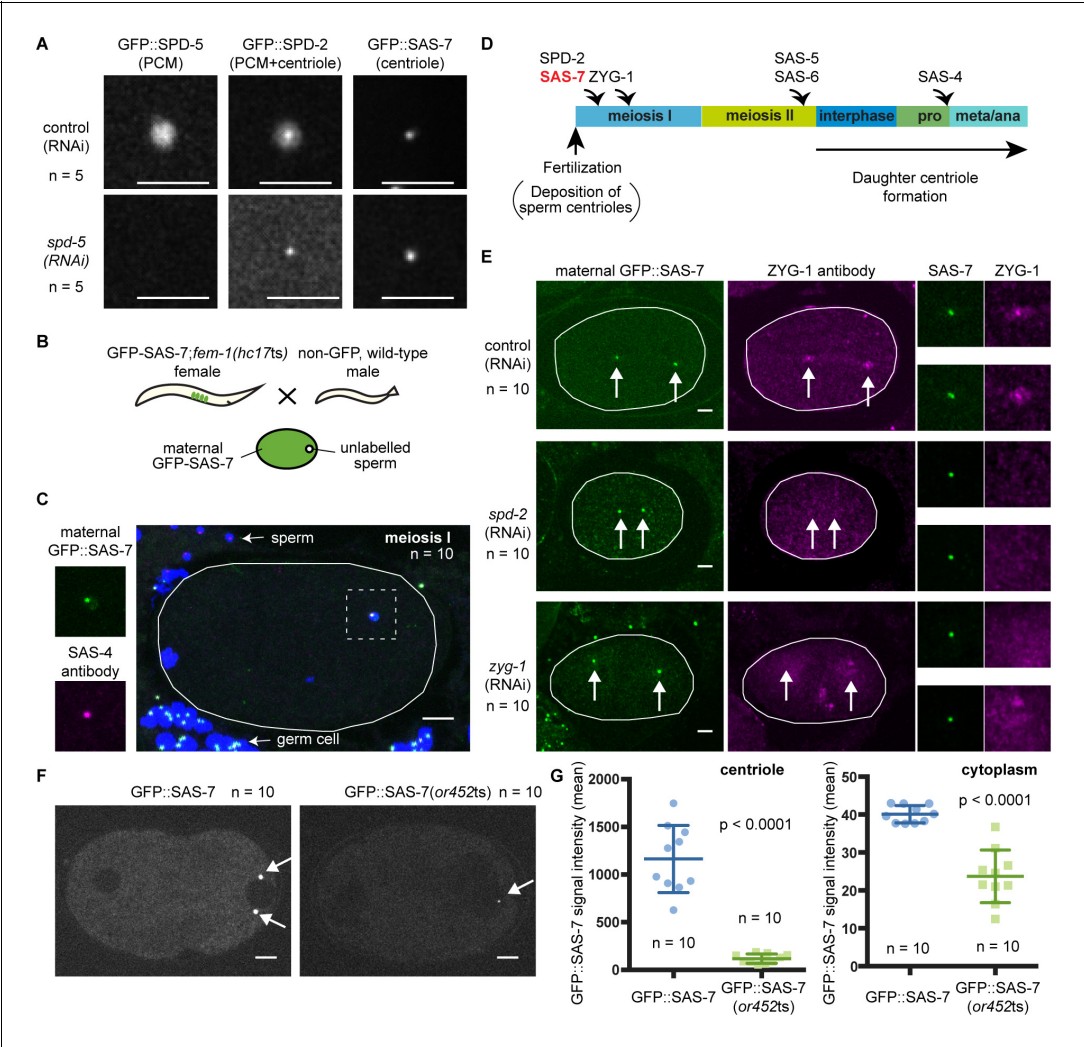

**Figure 5.** SAS-7 localizes to centrioles independently of SPD-2. (**A**) Comparison of GFP::SAS-7, GFP::SPD-5 and GFP::SPD-2 localization. SPD-5 is a PCM component. SPD-2 localizes to both the centriole and the PCM. SAS-7 localized exclusively to the centriole. (**B**) Schematic illustration of experiments performed in C, E and *Figure 5—figure supplement 1C*. (**C**) Localization of maternal SAS-7 protein to the site of daughter centriole formation. GFP::SAS-7 and SAS-4 antibody staining are shown in green and magenta. Note that most of the SAS-4 on centrioles at this stage is paternal protein (*Delattre et al., 2006*; *Pelletier et al., 2006*). (**D**) Schematic illustration of the timing of core protein recruitment to the centriole (*Delattre et al., 2006*; *Pelletier et al., 2006*; this study). (**E**) Maternal SAS-7 localization in *spd-2(RNAi)* and *zyg-1(RNAi)* embryos. Maternal SAS-7 and ZYG-1 antibody staining shown in green and magenta, respectively. ZYG-1 centriolar localization was lost in *spd-2(RNAi)* as reported previously (*Delattre et al., 2006*; *Pelletier et al., 2006*). Arrows indicate the position of centrioles. (**F**) GFP-tagged SAS-7(*or452*ts) showed reduced expression level compared to wild-type. (**G**) Quantified GFP::SAS-7 intensity on centriole (left graph) and cytoplasm (right graph). P-values were calculated by Welch's t-test. Scale bars indicate 5 μm.

The following figure supplement is available for figure 5:

**Figure supplement 1.** SAS-7 localizes to the centriole throughout cell division.

(*Figure 2—figure supplement 1A*, *Figure 5A*, *Figure 5—figure supplement 1A and 1B*). To determine if SAS-7 localizes to centrioles or to the surrounding PCM, we used RNAi to knock down SPD-5, which is essential for PCM formation, and then examined both SPD-2::GFP and GFP::SAS-7 in live embryos. SPD-2 is known to localize to both the PCM and the centriole. We found that centriolar SPD-2 appeared normal in *spd-5(RNAi)* embryos, but as expected PCM-associated SPD-2 was reduced (*Figure 5A*). The *spd-5(RNAi)* embryos showed centriolar foci of GFP::SAS-7 and SPD-2:: GFP that were similar in size and the GFP::SAS-7 signal was not altered (*Figure 5A*). We conclude that SAS-7 is a new centriole component essential for centriole duplication.

*C. elegans* centriole duplication can be monitored by specifically labeling maternally expressed centriolar proteins. Oocyte centrioles are eliminated before fertilization, and the sperm is the sole source of centrioles provided to the fertilized egg. Subsequently, the five core centriole components SPD-2, ZYG-1, SAS-5, SAS-6 and SAS-4 are sequentially recruited from the maternal cytoplasm to the site of daughter centriole formation (see Introduction), as documented by labeling maternal centriole components with fluorescent protein fusions and mating feminized transgenic hermaphrodites with wild-type (unlabeled) males (*Figure 5B*). Using this technique, we determined the timing of SAS-7 recruitment to the site of daughter centriole assembly during the first cell cycle after fertilization. We found that maternal GFP::SAS-7 was present at centrioles at meiosis I, just after fertilization (*Figure 5C* and *Figure 5—figure supplement 1C*). Thus SAS-7 localizes to centrioles as early as SPD-2, the earliest acting centriole assembly factor known (*Figure 5C and D*). Because SPD-2 is required for the centriolar localization of the other four core centriole components, we tested whether SAS-7 localization also requires SPD-2. Remarkably, neither *spd-2(RNAi)* nor *zyg-1(RNAi)* affected centriolar GFP::SAS-7 localization (*Figure 5E*). We conclude that SAS-7 localizes to centrioles independently of SPD-2.

We also inserted GFP coding sequences into the *sas-7* genomic locus of *sas-7(or452*ts) mutants using CRISPR/Cas9. We obtained a GFP::SAS-7(*or452*ts) line and detected the mutant protein at centrioles, although at greatly reduced levels both at centrioles and in the cytoplasm (*Figure 5F and G*). Thus the *or452*ts mutation likely results in reduced expression of a truncated protein.

## SAS-7 regulates SPD-2 centriolar localization

Consistent with SAS-7 acting early in centriole duplication, a genome-wide yeast two-hybrid screen identified SAS-7 as a SPD-2 binding protein (*Boxem et al., 2008*; *Li et al., 2004*), although SAS-7 function has otherwise not been investigated. We confirmed the interaction between SAS-7 and SPD-2 using a yeast two-hybrid assay and used a series of N-terminal and C-terminal deletions to determine that SAS-7 interacts with SPD-2 through its C-terminus (*Figure 6A*). We identified two SPD-2 binding domains in the C-terminus of SAS-7 (*Figure 6A*). Interestingly, SAS-7 constructs that lack the region deleted in *or452*ts allele lost one of the SPD-2 interacting domains, suggesting that the SAS-7/SPD-2 interaction is impaired but not completely lost in *sas-7(or452*ts) mutants (*Figure 6A*).

To examine the effect of the *sas-7(or452*ts) mutation on SPD-2 function, we quantified the total centrosomal SPD-2::GFP signal during the first and second embryonic mitotic cell divisions. We found that the SPD-2::GFP signal was reduced by more than half in *sas-7* mutants compared to control heterozygotes during prophase and interphase, while the signals at nuclear envelope breakdown (NEBD) were almost identical (*Figure 6B and C*). In addition, the SPD-2 signal started to increase later and decrease sooner than in the control (*Figure 6B and C*). Thus in *sas-7(or452*ts) mutants, the amount of SPD-2::GFP signal is reduced and the timing of SPD-2 incorporation (PCM growth) is altered.

As we observed reduced SPD-2::GFP in *sas-7(or452*ts) mutants at interphase and early prophase, when PCM recruitment is not prominent (*Figure 6B and C*), we hypothesized that SAS-7 regulates centriolar SPD-2 localization. To test this idea, we knocked-down SPD-5 using RNAi to remove the PCM and detected reduced levels of SPD-2::GFP in *sas-7(or452*ts); *spd-5(RNAi)* embryos compared to *spd-5(RNAi)* embryos (*Figure 6D*), consistent with SAS-7 regulating centriolar SPD-2 localization. Because SPD-2 regulates downstream core component localizations, we also examined SAS-6::GFP localization in *sas-7* mutants (*Figure 6E*). In wild-type embryos, SAS-6 is recruited to centrioles around prophase, whereas it fails to localize when SPD-2 is depleted by RNAi (*Delattre et al., 2006*; *Pelletier et al., 2006*). To see if SAS-7 plays a similar role in recruiting core centriole proteins, we examined SAS-6::GFP localization (*Figure 6E*). We found that SAS-6::GFP levels were reduced at

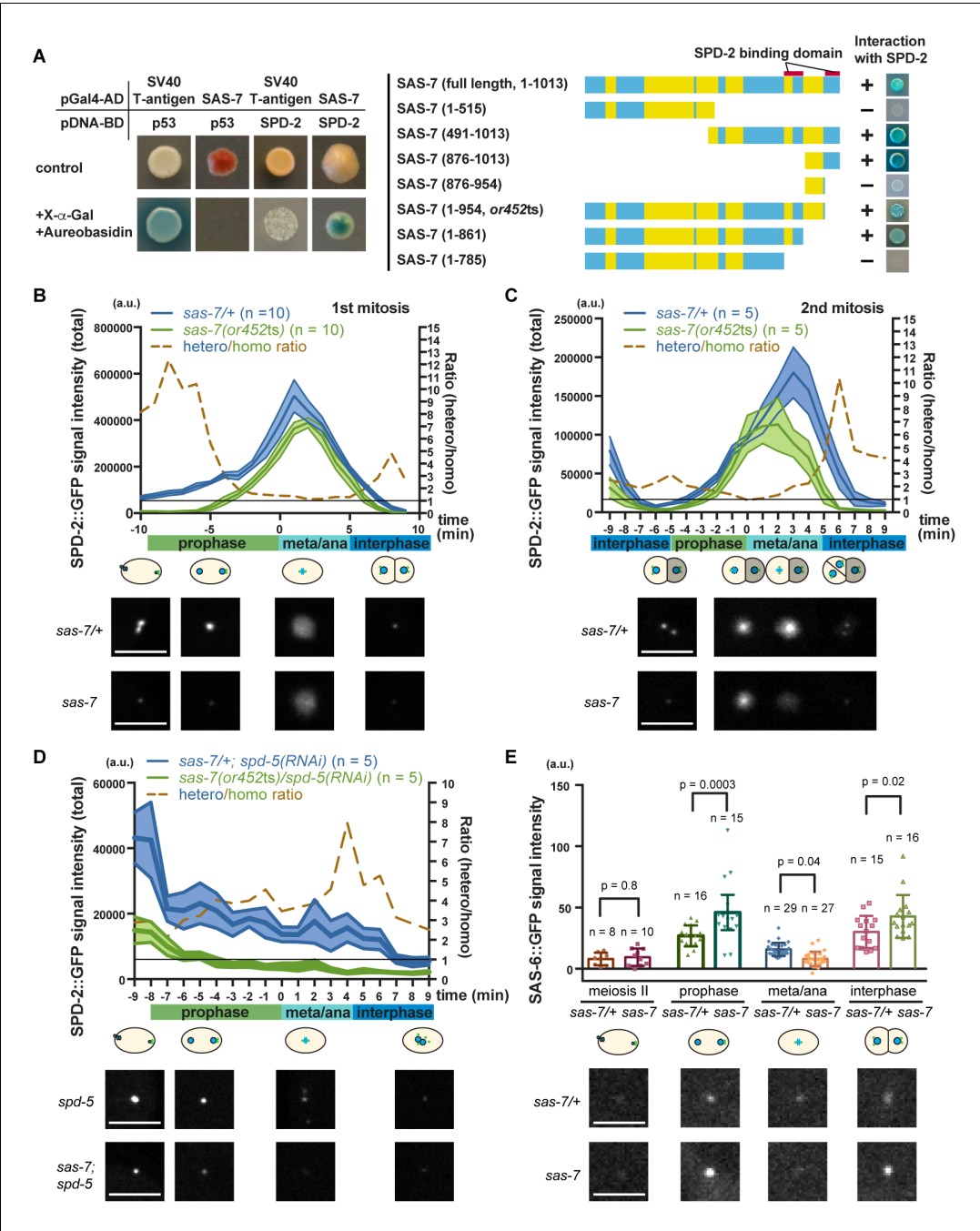

**Figure 6.** SAS-7 regulates SPD-2 centriolar localization. (**A**) Yeast two-hybrid assays to test the interaction between SAS-7 and SPD-2. In the left panels, positive control, negative control and experiment are shown from left to right. Yellow in the gene model represents the coiled-coil motifs. (**B, C**) Quantified SPD-2::GFP signal intensity in the first mitosis ($P_0$) and the second mitosis (AB) in B and C, respectively. (**D**) Quantified SPD-2::GFP signal in *spd-5(RNAi)* background. *sas-7(or452*ts) heterozygous siblings were used as a control in B-D. Mean ± SEM are shown. Times are relative to NEBD. (**E**) Quantified SAS-6::GFP signal intensity. Error bars indicate 95% confidence intervals (CI). P-values were calculated by one-way ANOVA with Holm-Sidak's multiple comparison test. Scale bars indicate 5 μm.

metaphase and anaphase, but were increased at prophase and interphase in *sas-7* mutants (*Figure 6E*). These seemingly paradoxical results might reflect the previous finding that SAS-6::GFP recruitment to, and incorporation into centrioles are separable steps, with unincorporated SAS-6::GFP being lost before metaphase and anaphase (*Lettman et al., 2013*). Our results suggest that SAS-6 recruitment may be up-regulated but that incorporation nevertheless fails in *sas-7(or452*ts). Taken together, these results suggest that SAS-7 regulates SPD-2 centriolar localization and affects the recruitment and incorporation of downstream core components.

## SAS-7 regulates PCM function and integrity

Some of the *sas-7(or452ts)* mutant embryos that formed bipolar spindles divided into equally-sized daughter cells, in contrast to the polarized divisions of wild-type embryos that result in unequally sized daughters, suggesting that the mutants were defective in embryonic polarity. Previous studies have shown that the centriole/PCM component SPD-2 and the PCM component SPD-5 are required to establish polarity in the 1 cell embryo, and hence the anterior-posterior (A-P) body axis (*Cowan and Hyman, 2004*) (*Figure 7A*). As other centriole core components have not been

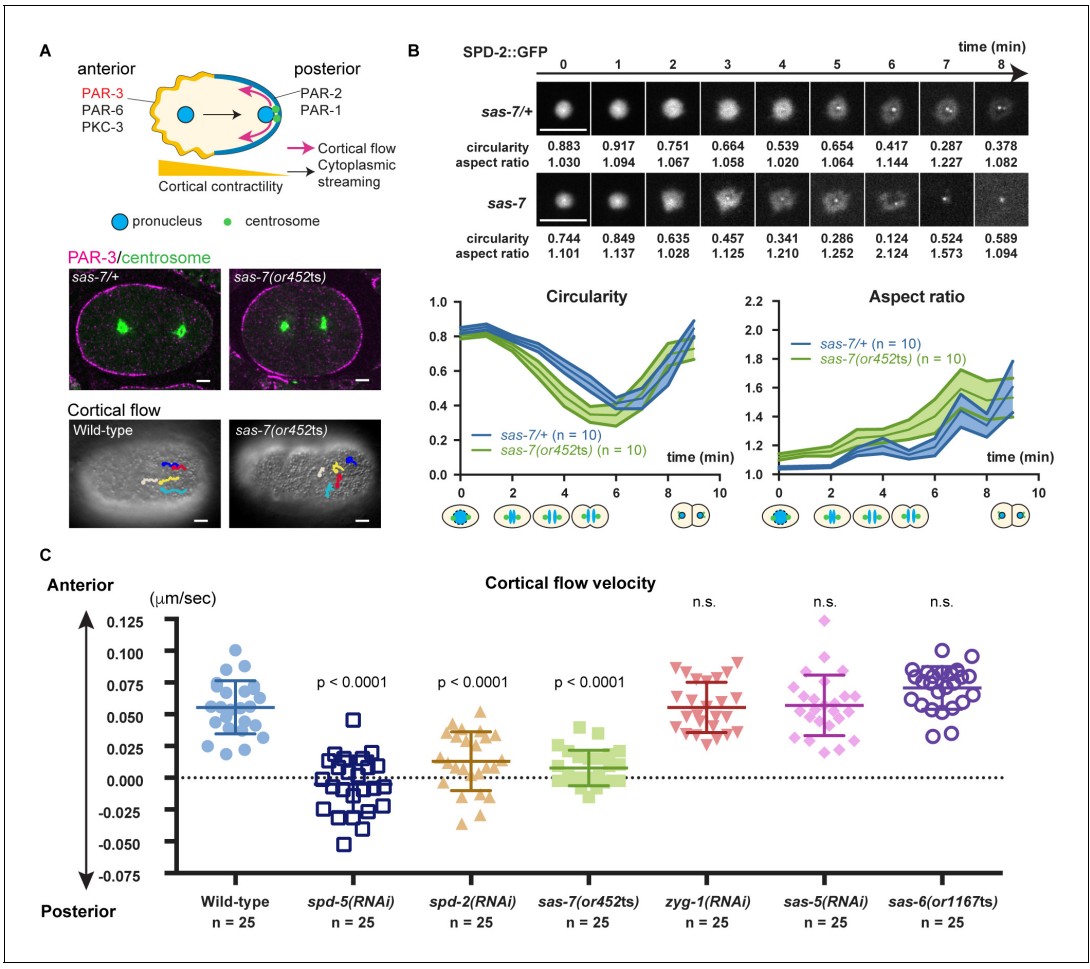

**Figure 7.** SAS-7 regulates PCM function and integrity. (**A**) Top: Schematic illustration of PAR-dependent cell polarity at 1 cell stage. Middle: PAR-3 and SPD-5 antibody staining at telophase. Cortical PAR-3 and centrosomal SPD-5 are shown in magenta and green, respectively. Bottom: Trajectories of surface yolk granule movements during 120 s in early prophase. (**B**) PCM shapes in *sas-7* mutant. Using SPD-2::GFP, PCM circularity and aspect ratios of anterior centrosomes were calculated and shown in the graphs. Time is relative to NEBD. *sas-7* heterozygous siblings were used for control. (**C**) Cortical flow velocities in control and mutant embryos. Cortical flows were calculated and anterior-directed movements given a positive value. Mean ± SD are shown. P-values were calculated by one-way ANOVA with Bonferroni's multiple comparison test. Scale bars indicate 5 μm.

reported to be involved in this process, the PCM and not the centriole is thought to polarize the A-P body axis.

After the initial establishment of A-P polarity, the polarity proteins PAR-3, PAR-6 and PKC-3 become restricted to the anterior half of the cell cortex while PAR-2 and PAR-1 localize to the posterior (*Munro and Bowerman, 2009*) (*Figure 7A*). Furthermore, non-muscle myosin also becomes enriched anteriorly, with the resulting asymmetry in cortical contractility along the A-P axis causing cytoplasm in the center of the zygote to flow posteriorly (cytoplasmic streaming), and cortical cytoplasm to move anteriorly (cortical flow) (*Mayer et al., 2010*; *Niwayama et al., 2011*) (*Figure 7A*). We found that anteriorly-directed cortical flow was absent in *sas-7(or452*ts) mutants and that PAR-3 polarity was often lost (*Figure 7A*; 50%, n = 8), confirming that SAS-7 is required for the proper establishment of the A-P axis of cell polarity at the one-cell stage. We also confirmed that depletion of the PCM components SPD-2 and SPD-5 resulted in loss of cortical flow (*Hamill et al., 2002*; *Cowan and Hyman, 2004*), while knock-down of the other centriolar components ZYG-1, SAS-5 and SAS-6 did not (*Figure 7C*).

The polarity defects in *sas-7(or452*ts) mutants suggest that SAS-7 regulates PCM function, along with SPD-5 and SPD-2. Two results further support this conclusion. First, PCM shapes were abnormal in *sas-7(or452*ts) mutants. In wild-type one-cell stage embryos, the PCM was circular with an aspect ratio near 1.0 at NEBD (*Figure 7B*). As mitosis progressed, circularity decreased and the aspect ratio increased until PCM disassembly at telophase (*Figure 7B*). However, in *sas-7* mutants the PCM underwent rapid deformation, suggesting that PCM integrity is abnormal (*Figure 7B*). Second, as shown in *Figure 6B and C*, PCM growth as assayed by SPD-2 recruitment was delayed *in sas-7* mutants. We conclude that SAS-7 is required not only for centriolar SPD-2 localization and centriole duplication, but also for proper PCM assembly and for the PCM to induce the A-P axis of cell polarity.

## Discussion

### SAS-7 acts upstream of SPD-2 in centriole duplication and PCM formation

We identified *C. elegans* SAS-7 as a new centriole component that is required for the daughter centriole to acquire centriole duplication competence and thus become a mother centriole, and for the maintenance of mother centriole duplication competence. SAS-7 localizes to the centriole independently of SPD-2, and regulates SPD-2 centriolar localization. Therefore, SAS-7 is the earliest acting centriole duplication protein yet identified (*Figure 8A*). SAS-7 also is required for proper centriole assembly, as *sas-7* mutant centrioles are shorter and have morphological defects in the peripheral structure we call the paddlewheel. Finally, like SPD-2, SAS-7 is required for proper PCM formation and for the PCM-dependent establishment of the A-P axis of cell polarity in one-cell stage embryos.

Both SAS-7 and SPD-2 have dual functions in PCM formation and centriole assembly (*Figure 8A*), but only SPD-2 is detectable in the PCM (*Figure 5A*). In this respect SAS-7 is similar to SAS-4, which localizes to centrioles but appears to regulate PCM formation (*Kirkham et al., 2003*). Because SAS-7 can bind SPD-2 in a yeast two-hybrid assay, it is possible that SAS-7 directly recruits SPD-2 to the PCM (*Figure 8A*, arrow 1). Alternatively, the loss of centriolar SPD-2 in *sas-7* mutants might result in the unstable localization of SPD-2 to the PCM and hence cause defects in PCM integrity and function (*Figure 8A*, arrow 2). Consistent with this model, it has recently been proposed that centrioles regulate PCM growth by catalyzing the conformational change of a centrosomal component to an assembly competent form (*Zwicker et al., 2014*). The loss of centriolar SPD-2 in *sas-7* mutants might impair such a catalytic activity of the centriole and thereby affect the dynamics and integrity of the PCM. Finally, although we observed a substantial reduction in SPD-2 centriolar levels, SPD-2 still localized to the centriole in *sas-7(or452*ts) mutants (*Figure 6*). While there may be additional regulators of SPD-2 centriolar localization, the *sas-7(or452*ts) allele expresses residual SAS-7 mutant protein. Therefore, the partial penetrance of the monopolar mitotic spindle phenotype in *sas-7(or452*ts) mutant embryos as well as residual SPD-2 centriolar localization might reflect an incomplete loss of SAS-7 function at the restrictive temperature.

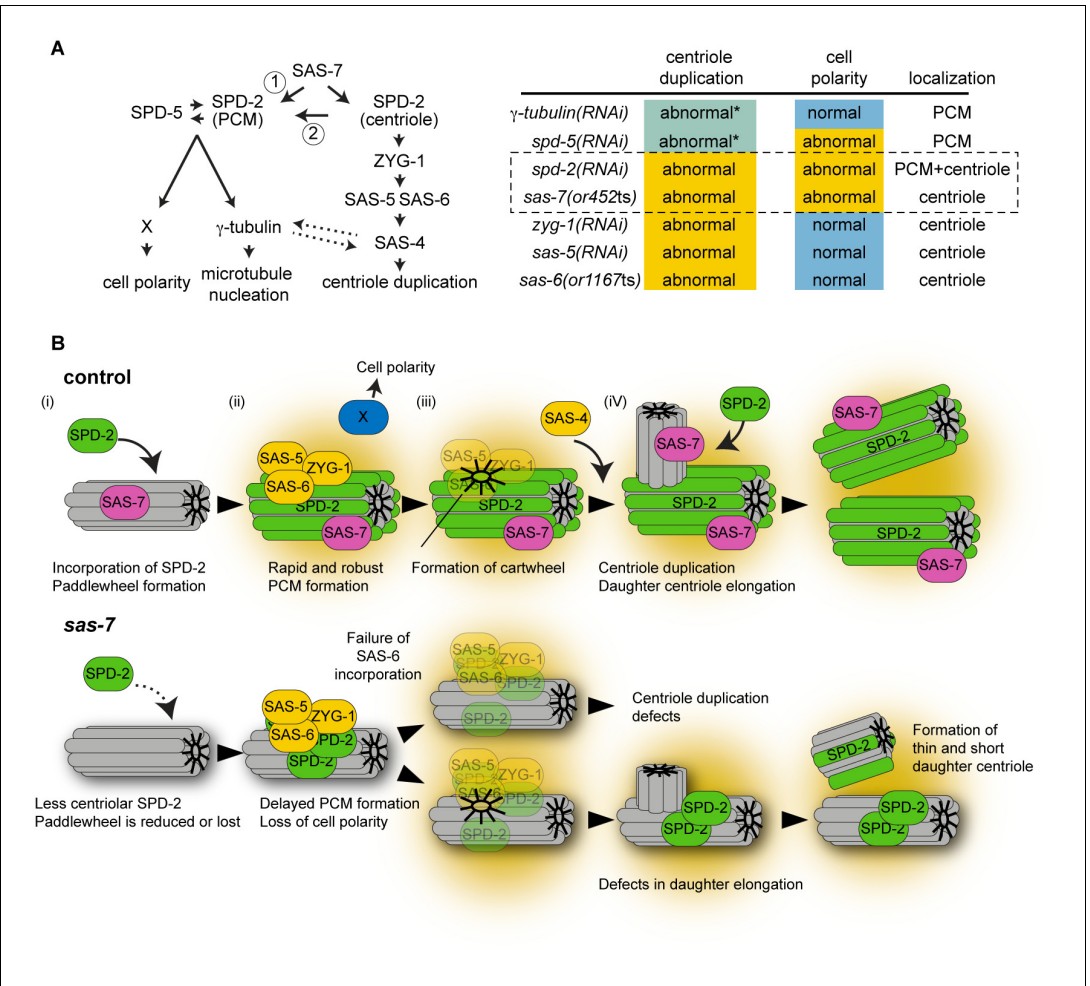

**Figure 8.** SAS-7 acts upstream of SPD-2 and is critical for both centriole assembly and PCM formation. (**A**) Left: Proposed SAS-7 pathway regulates both centriole biogenesis and PCM formation. SAS-7 may directly regulate PCM SPD-2 (arrow 1), or indirectly affect a catalytic activity of the centriole (arrow 2; *Zwicker et al., 2014*) that controls PCM growth (see text). An inducing factor for cell polarity, X, is unknown. SAS-4 is reported to regulate PCM size (*Kirkham et al., 2003*) while PCM component(s) regulate proper SAS-4 recruitment (*Dammermann et al., 2004*) (dotted arrows). Right: A summary of centriole duplication defects and cell polarity defects in the mutants of PCM and centriole components (*Gönczy, 2012*; this study). *Unlike centriole duplication-defective and some partial loss of function *sas-7(or452*ts) mutants, depletion of PCM components γ-tubulin and SPD-5 does not fully block centriole duplication. (**B**) A model for SAS-7 function during centriole biogenesis and PCM formation. (i) SAS-7 recruits SPD-2 to the centriole to regulate the formation of the paddlewheel. (ii) SPD-2 recruits ZYG-1/SAS-5/SAS-6 to initiate daughter centriole formation. SPD-2 also promotes PCM formation to induce cell polarity by an unknown factor X. (iii) Cartwheel formation. (iv) Centriole duplication and SAS-7 recruits SPD-2 to the daughter centriole to form paddlewheel.

## Human Cby2 and CFAP57 have weak homology to SAS-7

The five previously identified core centriole duplication proteins in *C. elegans* are widely conserved across animal phylogeny, yet they are highly diverged from their homologs. We have not been able to detect proteins outside of the nematode phylum that are clearly homologous to SAS-7. However, the N-terminal and C-terminal regions of SAS-7 exhibit weak homology to the human SPERT/Cby2 and CFAP57/WDR65 proteins, respectively (*Figure 1—figure supplement 3*, *Figure 1—figure supplement 4*). Previous RNAi knock-down of Cby2 and CFAP57 in human cells did not cause defects in centriole duplication (*Balestra et al., 2013*), and unlike CFAP57 family proteins, SAS-7 does not contain WD40 repeats (*Figure 1—figure supplement 4*). The human Cby2 paralog Cby1 and a fly

ortholog called Chibby also are not required for cell division but are required for basal body formation (*Enjolras et al., 2012*; *Lee et al., 2014*), consistent with a possible role in centriole function. Thus, it is possible but inconclusive that SAS-7 homologs exist outside of nematodes.

## New ultrastructural insights into *C. elegans* centriole architecture and assembly

Pioneering ultrastructural work by *Pelletier et al. (2006)* reported that *C. elegans* centrioles were 172 nm long and 110 nm in diameter. We have found the length of centrioles to be essentially the same—185 nm. However, our TEM results indicate that centrioles are wider than previously reported. Specifically, we describe a peripheral centriolar structure we call the paddlewheel that decorates the outer edges of the centriole and makes the centriole diameter 175 ± 9.05 nm (*Figure 4*). In our samples the centriole minus the paddlewheel is approximately 114 nm in diameter, consistent with previous reports (*O'Toole et al., 2003*; *Pelletier et al., 2006*). We also describe a cartwheel like structure: an 18 nm inner tube with an electron-lucent center and spokes emanating from it. We suggest that our fixation procedures, which are different from Pelletier et al. (see Materials and methods), may account for our ability to better detect these ultrastructural features of the *C. elegans* centriole.

Structural evidence that SAS-6 family members can oligomerize to form a cartwheel with 9-fold radial symmetry, and thereby provide a template for the assembly of the 9-fold radially symmetric centriole microtubules, is central to our current understanding of centriole duplication. Our ultrastructure analysis now provides evidence that the *C. elegans* centriole also has a cartwheel-like structure, as others have predicted (*Gönczy, 2012*; *Hilbert et al., 2013*). Although a central hub-like inner tube was evident in our TEM images, spokes that emanate from a central hub were not obviously present. Nevertheless, electron density possibly related to central hub spokes was detected in some images (*Figure 4B*). The ambiguity in our detection of spokes is consistent with a previous analysis suggesting that *C. elegans* SAS-6 may form a spiral structure rather than the stacks of aligned cartwheels proposed to template centriole microtubules in other species (*Hilbert et al., 2013*). Thus the *C. elegans* SAS-6 spokes may not consistently align to generate the electron density corresponding to the spokes that have been clearly observed in TEM images of centrioles from other eukaryotes (*Guichard et al., 2012*).

Our ultrastructure images indicate that peripheral structures, possibly related to electron density referred to previously as appendages (*Pelletier et al., 2006*), span the entire length of the centriole and depend on SAS-7 function. We propose to rename this structure the paddlewheel to distinguish it from vertebrate appendages that are found only at the distal end of mother centrioles and are dispensable for cell division (*Paintrand et al., 1992*; *Vorobjev and Chentsov, 1982*; *Ishikawa et al., 2005*). Electron dense structures that, like the *C. elegans* paddlewheel, extend along the entire centriole length have also been observed in *Drosophila* (*Callaini et al., 1997*) and possibly in human cells (*Paintrand et al., 1992*), although their function and molecular requirements are unknown.

Our imaging of the paddlewheel also revealed a chirality for the *C. elegans* centriole that has not been reported previously. In other eukaryotes, centrioles are coated by doublet or triplet microtubules that are twisted in a clock-wise direction when viewed from the distal end (*Geimer and Melkonian, 2004*; *González et al., 1998*; *Paintrand et al., 1992*). Because the *C. elegans* centriole is coated by singlet microtubules, any chirality of the centriole cannot be determined from the arrangement of its microtubules. However, the individual paddles within the paddlewheel structure we describe here also were twisted in a clock-wise direction when viewed from the distal end, documenting a remarkable conservation of centriole chirality across phyla.

Interestingly, SAS-7 is required both for the formation of the paddlewheel and for SPD-2 centriolar recruitment. Thus it is possible that the paddlewheel may include SPD-2 protein. Consistent with this idea, immuno-EM staining indicated that SPD-2 protein is located about 100 nm away from the center of centriole (*Pelletier et al., 2004*), corresponding to the position of the paddlewheel in our TEM images. Super resolution microscopy (3D-SIM) of fly SPD-2 (*Mennella et al., 2012*) and its human homolog Cep192 (*Lawo et al., 2012*) also has shown that these proteins localize just outside of centriole microtubules. Thus the paddlewheel structure we have described may represent a conserved centriole structure where SPD-2 family members are localized. Consistent with this idea, the possibly related peripheral structures described previously appear to be absent in a *spd-2(RNAi)* centriole, although this difference was not discussed (*Pelletier et al., 2004*). In future studies, both

the existence of a cartwheel-like structure, and a SPD-2 requirement for paddlewheel formation, need to be quantitatively assessed in multiple *sas-6* and *spd-2* mutant embryos, respectively.

Finally, although we did not observe any defects in microtubule number in *sas-7(or452*ts) centrioles, or in central tube and inner tube formation (*Figure 4—figure supplement 1B*), we did observe defects in centriole elongation (*Figure 4D*). In mammalian cells, over-expression of the SAS-4 related protein CPAP increases centriole length (*Kohlmaier et al., 2009*; *Schmidt et al., 2009*; *Tang et al., 2009*). Moreover, a recent study suggests that CPAP regulates centriole length, with MBD and LID domains promoting and limiting centriole microtubule elongation, respectively (*Sharma et al., 2016*). Therefore, the short centriole phenotype in *sas-7* mutant could be due to abnormal SAS-4 function, which is consistent with the predicted SAS-7-SAS-4 interaction reported by a yeast two-hybrid screen (*Boxem et al., 2008*). Furthermore, our data suggests that paddlewheel formation and centriole elongation might be mechanistically linked, as both are defective in *sas-7* mutants.

## Model for SAS-7 function in PCM formation and centriole duplication

In mammalian cells, the daughter centriole acquires competence for both centriole duplication and PCM recruitment during mitosis; this phenomenon is called centriole-to-centrosome conversion (*Wang et al., 2011*). Centriole duplication and PCM formation defects in *sas-7* mutants suggests that this process may be conserved in *C. elegans*, but the dual functions of SPD-2 in centriole duplication and PCM assembly also might account for both phenotypes. Therefore, we propose a model for the SAS-7 function that focuses on SPD-2 protein activity without referring to the mammalian process of centriole-to-centrosome conversion.

Based on our analysis of the *sas-7* mutant phenotype, we propose that SAS-7 recruits SPD-2 to centrioles to build the paddlewheel structure (*Figure 8B*). In the absence of SAS-7, SPD-2 localization is reduced, resulting in delayed PCM growth that in turn leads to a loss of cell polarity (*Figure 8B*). The reduced SPD-2 level also results in a failure of SAS-6 incorporation to form the cartwheel-like spiral structure and thus a failure in centriole duplication (*Figure 8B*). However, once the paddlewheel is formed, the centriole may become duplication-competent without SAS-7 being required further for the duplication process, although SAS-7 may be required for longer term maintenance of duplication competence (*Figure 3*). In cases where centrioles are duplicated in *sas-7* mutants, due to incomplete loss of SAS-7 function or the presence of other partially redundant factors, newly formed daughter centrioles again cannot recruit SPD-2 protein efficiently and have defects in paddlewheel formation and centriole elongation (*Figure 8B*). This model can explain how SAS-7 functions as an organizer of centriole duplication, centriole assembly and robust PCM formation, and accounts both for the incomplete penetrance we observed in centriole duplication and cell polarity, and for the more fully penetrant defects in centriole elongation and paddlewheel structure.

## Materials and methods

### Worm strains and maintenance

*C. elegans* strains were maintained on NGM agar plates and fed *E. coli* following standard protocols (*Brenner, 1974*). Temperature sensitive strains were kept at 15°C for continuous propagation but were shifted in L1 stage to 26°C for genetic test and to observe the mutant phenotype of their offspring. The following alleles were used, *sas-7(or452*ts), *sas-7(or1942)*, *sas-7(or1945)*, *sas-7(tm1105)*, *sas-6(or1167*ts), *fem-1(hc17*ts), *him-8(e1489)*, *nDf40*, *eT1*. The following transgenes were used, *sas-7 (or1940)* (GFP::SAS-7), *sas-7(or1953)* [GFP::SAS-7(*or452*ts)], *ddIs299* (YFP::SPD-5, a gift from Tony Hyman), *ddIs195* (SPD-2::GFP, a gift from Tony Hyman), *isIs21* (SAS-6::GFP, a gift from Pierre Gönczy). *or1942*, *or1945*, *or1940* and *or1953* were generated by CRISPR/Cas9 mediated genome editing as described below.

### Gene identification: mapping and transformation rescue

Three-factor mapping using visible markers and snip-SNPs (*Wicks et al., 2001*) was done to narrow the chromosomal position of the gene responsible for the *or452*ts phenotype. Fosmids spanning this region of the *C. elegans* genome were prepared from the Source BioScience fosmid library following instructions from the manufacturer followed by DNA minipreps (Qiagen, Venlo, Netherlands).

Purified fosmid DNA was digested with NotI and injected along with carrier DNA and transformation markers into mutant worms. The carrier used was HB101 *E. coli* DNA purified using a GeneElute Bacterial Genomic DNA kit (Sigma-Aldrich, St. Louis, MO), which was then digested with PvuII and purified on a QiaQuick column (Qiagen). The linearized markers for transformation used were Plasmid 19327: pCFJ90-Pmyo-2::mCherry::unc-54utr (Addgene) and pRF4-rol-6. Worms exhibiting red rolling marker phenotypes were scored for rescue of the embryonic lethality.

## Whole-Genome sequencing and data analysis

Genomic DNA extraction was performed using a DNeasy kit (Qiagen), using sterile L1s after hypochlorite treatment as input. Libraries were prepared for sequencing using Nextera DNA Library Preparation kit (Illumina, San Diego, CA), following the included protocol with no modifications. Sequencing was performed on an Illumina HiSeq 2500 at the University of Oregon's Genomics Core Facility.

All data analysis tools were run using the Galaxy platform (*Giardine et al., 2005*; *Goecks et al., 2010*). The workflow used is the same as described (*Minevich et al., 2012*), with some modifications. Read mapping was performed using Bowtie2 (*Langmead and Salzberg, 2012*), alignment quality control was performed using a combination of Picard (http://broadinstitute.github.io/picard/) and GATK (*DePristo et al., 2011*), and variant detection and analysis was performed using a combination of GATK, CloudMap, snpEff (*Cingolani et al., 2012*), and Bedtools (*Quinlan and Hall, 2010*). SNP mapping plots were generated using the CloudMap tool.

## cDNA cloning

Total RNAs from mixed-stage wild-type, *sas-7(tm1105)* and *sas-7(1940)* (GFP::SAS-7 strain) were isolated by TRIzol LS reagent (Life Technologies, Eugene, OR). cDNAs were amplified using SuperScript One-step RT-PCR kit (Invitrogen).

## CRISPR/Cas9-mediated genome editing

CRISPR/Cas9-mediated genome editing was performed as described (*Dickinson et al., 2013*; *Kim et al., 2014*) for two purposes: 1) to insert GFP in frame at N-terminal of *sas-7*/T07C4.10, which allowed for visualization of the endogenous protein, and 2) to generate mutants of *sas-7* to confirm the *or452*ts phenotype. For *gfp::sas-7* repair template for homologous recombination, a 2.3 kb fragment (chr III: 10319806–10322122) amplified from genomic DNA was cloned into the pJET1.2 vector (ThermoFisher Scientific, Whaltham, MA). The GFP-coding region amplified from pSO26 (*O'Rourke et al., 2007*) was inserted at the N terminus of *sas-7* by Gibson assembly (*Gibson et al., 2009*). A point mutation was introduced to a protospacer adjacent motif (PAM) of small guide RNA (sgRNA) targeting site (ggctttaaaatcaactcaccg<u>tgg</u> to ggctttaaaatcaactcaccg<u>tag</u>) by inverse PCR. *eft-3*p::Cas9 plasmid with *sas-7* N terminus targeting sgRNA was made from pDD162 plasmid (Addgene) (*Dickinson et al., 2013*). The *gfp::sas-7* repair template, Cas9/sgRNA plasmid, and selection marker pRF4 were then injected into wild-type or *sas-7(or452*ts) young adults. The Roller animals carrying pRF4 were screened for GFP expression as described (*Kim et al., 2014*). For the *sas-7* mutants, sgRNA targeting ggctttaaaatcaactcaccgtgg and cggtgttaaatctctactggagg were inserted into pDD162 to generate *or1942* and *or1945*, respectively and injected with *unc-22* sgRNA plasmid (pCCM935, a gift from Craig Mello) as described (*Kim et al., 2014*). In or1942 mutants, TGAGT sequence starting from the second residue of intron 1 (chr III: 10321123–10321127) was deleted and adenine was inserted, resulting in the loss of splicing donor site. In or1945, CTGGA sequence starting from the 18th residue of exon 3 (chr III: 10321425–10321429) was deleted and result in the formation of premature stop codon at 104 a.a (glutamine in the wild-type).

## RNA interference

RNAi was done by feeding L1-L2 stage worms with HT115 *E. coli* expressing double stranded RNA (dsRNA) as described (*Kamath et al., 2001*). Bacteria containing the empty L4440 vector was used as negative control.

## Homology analysis

Conventional BLAST analysis with full-length SAS-7 sequence did not detect any homologous proteins outside the nematode phylum. Therefore, we partitioned SAS-7 sequences into 167 a.a. (500 bp) units that were used for PSI-BLAST analysis (NCBI, iteration 3) and detect Cby2/SPERT. After yeast-two hybrid experiments, we also performed the PSI-BLAST analysis using the C-terminal SAS-7 sequence containing SPD-2-binding domains (786–1013 a.a.) and detected CFAP53/WDR65. Multiple alignment was performed with Clustal Omega (*Sievers et al., 2011*) and processed with Jalview (*Waterhouse et al., 2009*). Coiled-coil motifs in *Figure 6A* schematic drawings were obtained using COILS (*Lupas et al., 1991*).

## Statistical analysis

No statistical method was used to predetermine sample size. Reproducibility was confirmed in more than two independent experiments.

## Light microscopy

For differential interference contrast (DIC) images in *Figure 1* and *Figure 7*, images were captured with an Evolution LC camera controlled by Image Pro software and with a generic CCD camera controlled by ImageJ (*Schneider et al., 2012*), respectively. For immunofluorescence images in *Figure 1*, images were captured with an Olympus Fluoview scanning laser confocal microscope controlled by Fluoview software. For the live-imaging in *Figure 5A*, images were captured with a confocal unit CSU10 (Yokogawa electric, Musashino, Japan) and an EMCCD camera Image EM (Hamamatsu photonics, Hamamatsu, Japan) mounted on an inverted microscope Leica DMI 4000 (Leica Microsystems, Wetzlar, Germany) controlled by Metamorph (Molecular Devices, Sunnyvale, CA). For the fluorescence images in *Figure 5C*, images were captured with a LSM 700 confocal microscope system (Carl Zeiss, Oberkochen, Germany). For the rest of experiments, images were captured with a confocal unit CSU-W with Borealis (Andor Technology, Belfast, Northern Ireland) and an EMCCD camera iXon Ultra 897 (Andor Technology) mounted on an inverted microscope Leica DMi8 (Leica Microsystems) controlled by Metamorph. Note that *Figure 6B and D* were acquired with the same microscope setting, however, in *Figure 6C*, laser power was reduced to 33%. Images in *Figure 5*, *Figure 6E*, *Figure 5—figure supplement 1A and C* were obtained with 0.5 μm Z spacing, while the rest of images in *Figures 6* and *7* were obtained with 1 μm Z spacing. For the live imaging of SPD-2::GFP and SAS-6::GFP, images were taken every 1 min.

## Image quantification

All quantifications were performed with Fiji (RRID:SCR_002285; *Schindelin et al., 2012*). For the quantification of TEM images, the sizes of centriole structures were measured manually. GFP::SAS-7 signal intensity was quantified by using the region defined by the Yen threshold using the most central single plane that contained the centriole at prophase of $P_0$ cell. For SPD-2::GFP, the signal intensity, circularity and aspect ratio in the similarly defined area of centriole/centrosome were quantified for each time point. SAS-6::GFP signal was quantified using the same region as above but only the first time points were used for quantification to avoid photo bleaching. Cell division timing was determined relative to NEBD or cytokinesis. For the quantification of cortical flow in *Figure 7C*, the movements of five cortical yolk granules in five different embryos were tracked for at least 30 s by the Manual Tracking plug-in and calculated the velocity for each genotype.

## Immunofluorescence

Immunofluorescence was done by fixing and permeabilizing embryos as described (*Severson et al., 2000*). For the experiments in *Figure 5*, embryos were fixed for 3 min in −20°C methanol. Primary antibodies include anti-tubulin (RRID:AB_1904178; DM1A, Sigma-Aldrich, St. Louis, MO) used at 1:500, anti-SPD-5 (*Hamill et al., 2002*) used at 1:2000, anti-SAS-4 at 1:500 (a gift from Kevin O'Connell)(*Song et al., 2008*), anti-SAS-4 at 1:2000 (a gift from Pierre Gönczy), anti-ZYG-1 at 1:1000 (a gift from Pierre Gönczy) and anti-PAR-3 at 1:200 (RRID:AB_528424; P4A1, Developmental Studies Hybridoma Bank (*Nance et al., 2003*)). Fluorescently tagged secondary antibodies were purchased from Jackson ImmunoResearch (West Grove, PA) or ThermoFisher Scientific and used at 1:200 or

1:1000 following the manufacturers' instructions. Propidum Iodide or DAPI (ThermoFisher Scientific) was used to stain DNA.

## Transmission electron microscopy

For electron microscopy, synchronous populations of *sas-7(or452*ts) mutant or wild-type were grown at 15℃, then transferred as young adults to 26℃ for 10 hr. The adults were collected and bleached to release eggs following standard protocols (*Stiernagle, 2006*). Eggs were washed in egg buffer [5 mM Hepes 6.9, 110 mM NaCL, 4 mM KCl, 5 mM MgCl$_2$, 5 mM CaCl$_2$] then incubated in about 5 units of chitinase (Sigma-Aldrich C6137) in egg buffer until pretzel-staged embryos appeared round; these embryos have lost the chitin-containing layer of the eggshell, but remain surrounded by an inner layer. The inner layer was removed by mechanical shearing; the embryo slurry was backloaded into a drawn Pasteur pipet, then gently expelled. The embryos were collected and fixed for 2 hr in 3% glutaraldehyde (Electron Microscopy Sciences, Hatfield, PA), 0.05 M sodium cacodylate (pH 7.0), and 2 mM MgCl$_2$. After rinsing with 3% sucrose in 0.05 M sodium cacodylate (pH 7.0), the embryos were postfixed in 1% Osmium (Electron Microscopy Sciences), 0.1 M cacodylate (pH 7.0) and 0.8% potassium ferricyanide (Sigma-Aldrich) for 45 min at 4℃. The embryos were rinsed with 0.05 M cacodylate (pH 7.0), and then stained with 0.2% tannic acid (Mallinckrodt, Dublin, Ireland) for 10 min. After rinsing with dH$_2$O, selected early embryos were collected, clustered on a pad of 1.5% bacto-agar (Difco (BD), Frankline Lakes, NJ), then covered with a top layer of agar. The sandwiched embryos were dehydrated and embedded with Epon resin (Electron Microscopy Sciences) following standard procedures (Specimen Preparation Protocol at http://sharedresources.fredhutch.org/training/electron-microscopy-procedures-manual). Serial sectioning was at 70–100 nm beginning just above, and ending just below, the nuclei. Sections were examined with a JEOL JEM 1400 Transmission electron Microscope (JEOL USA, Peabody, MA). To orient cross-sectional or lateral views of centrioles, sections were rotated and/or tilted using a dual-axis tomography grid holder (Fischione Instruments, Export, PA). Photography was with a 2K x 2K Gatan Ultrascan 1000XP camera (Gatan, Pleasanton, CA).

## Yeast two-hybrid

Yeast two-hybrid assays were performed with Matchmaker Gold Yeast Two-hybrid System (Clontech (Takara-Bio), Otsu, Japan) according to manufacturer's instruction. Full length and truncated versions of *sas-7*/T07C4.10 cDNA and SPD-2 full-length cDNA were inserted into pGADT7 AD vector and pGBKT7 DNA-BD vector, respectively. pGADT7-SV40 T antigen and pGBKT7-p53 were used for both negative and positive controls. Growth of transformed yeast strains was tested on SD/-Leu/-Trp plate for control and SD/-Leu/-Trp plate supplemented with X-α-Gal and 200 ng/ml Aureobasidin A for experiments. When two constructs interact, white and blue colonies appear on the control and experiment plates, respectively. When an interaction is negative, a red colony grows on the control plate while no colonies survive on experiment plates. Coiled-coil motif in *Figure 6A* schematic drawings were obtained using COILS (*Lupas et al., 1991*).

## Acknowledgements

We thank Steven MacFarlane and the FHCRC Electron Microscopy Resource for technical assistance, the Caenorhabditis Genetics Center (supported by National Institute of Health Infrastructure Programs; P40 OD010440), the National Bioresource Project, Shohei Mitani and Tony Hyman for strains, Pierre Gönczy and Kevin O'Connell for antibodies, Chris Doe for sharing lab equipment, Craig Mello for CRISPR reagents, Daiju Kitagawa, Brian O'Rourke and Meng-Fu Bryan Tsou for helpful discussions, and a critical reading of manuscript drafts by Pierre Gönczy, the editors Anna Akhmanova and Karen Oegema, and the reviewers. This work was supported by the NIH grant R15 GM071393-1 to DRH, R01 GM049869 to BB, R01 GM107474 to JRP, Human Frontier Science Program (LT000345/2012 L) and Journal of Cell Science Travelling Fellowship (JCSTF-141213) to KS.

# Additional information

## Funding

| Funder | Grant reference number | Author |
| --- | --- | --- |
| Human Frontier Science Program | LT000345/2012-L | Kenji Sugioka |
| Journal of Cell Science Traveling Fellowship | JCSTF-141213 | Kenji Sugioka |
| National Institute of General Medical Sciences | | Kenji Sugioka<br>Danielle R Hamill<br>Joshua B Lowry<br>Marie E McNeely<br>Molly Enrick<br>Alyssa C Richter<br>Lauren E Kiebler<br>James R Priess<br>Bruce Bowerman |
| National Institutes of Health | R15 GM071393-1 | Danielle R Hamill |
| National Institutes of Health | R01 GM107474 | James R Priess |
| National Institutes of Health | R01 GM049869 | Bruce Bowerman |

The funders had no role in study design, data collection and interpretation, or the decision to submit the work for publication.

## Author contributions

KS, DRH, Conceptualization, Data curation, Formal analysis, Funding acquisition, Investigation, Methodology, Writing—original draft, Writing—review and editing; JBL, MEM, ME, ACR, LEK, Data curation, Investigation; JRP, Conceptualization, Data curation, Formal analysis, Funding acquisition, Investigation, Methodology, Writing—original draft, Project administration, Writing—review and editing; BB, Conceptualization, Supervision, Funding acquisition, Project administration, Writing—review and editing

## Author ORCIDs

Kenji Sugioka, http://orcid.org/0000-0002-5830-9639
Bruce Bowerman, http://orcid.org/0000-0002-6479-8707

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
