## [Decision Letter]

Thank you for submitting your article "Centriolar SAS-7 acts upstream of SPD-2 to regulate *C. elegans* centriole assembly and PCM formation" for consideration by *eLife*. Your article has been reviewed by three peer reviewers, and the evaluation has been overseen by a Reviewing Editor and Anna Akhmanova as the Senior Editor. The following individuals involved in review of your submission have agreed to reveal their identity: Karen Oegema (Reviewer #1).

The reviewers have discussed the reviews with one another and the Reviewing Editor has drafted this decision to help you prepare a revised submission.

Summary:

In this manuscript the authors describe a novel centriolar protein SAS-7, which they show localizes to centrioles and is required for assembly of a new centriolar substructure that they call the paddle wheel. They further show that SAS-7 interacts with the previously characterized centriolar protein SPD-2. SPD-2 recruitment is abnormal in a temperature-sensitive mutant of SAS-7 that lacks a domain that they show is able to bind to SPD-2. This mutant also exhibits defects in the establishment of polarity, a process previously shown to require SPD-2. The mutant also results in the production of sperm that contain only a single functional centriole, and to low frequency defects in centriole duplication in embryos. SAS-7 is a clearly an interesting protein and all three reviewers were generally enthusiastic about the work. However, the reviewers also identified a number of issues that need to be addressed prior to consideration of a revised manuscript.

Essential revisions:

1) Whether or not SAS-7 makes a significant contribution to centriole duplication, and in what contexts (male germline vs. mitotic embryos) needs to be clarified. The prominent phenotype in the *or452*ts mutant is the presence of monopolar first embryonic divisions, which results from the production of sperm that have only one centriole. This phenotype could arise from a defect in centriole duplication in the male germline, or from another problem related to loss of the paddlewheel that limits the ability of centrioles to be packaged into sperm. More work monitoring centriole duplication and fate in the male germline would be needed to sort this out if the authors want to retain the focus of the paper on a role for SAS-7 in centriole duplication. Whether SAS-7 controls centriole duplication during the mitotic divisions in the embryo is a separate question from its role in the male germline. From the data presented, the role in centriole duplication during mitotic divisions appears to be quite minimal. If the role of SAS-7 in centriole duplication in embryos is relatively minor, the manuscript (and title) should be rewritten to focus the roles of SAS-7 in assembly of the paddle wheel, in controlling SPD-2 recruitment, and in controlling the ability of the centrosome to communicate centrosome position to the cortex to establish embryonic polarity.

Clarification would also include addressing the points raised in the following comment from reviewer #2:

"To clarify the extent to which the *or452* mutant affects maternal and paternal duplication it is important to clearly indicate at what developmental stage the *or452* hermaphrodites are shifted to the restrictive temperature. Table 1, footnote b specifies L1-L4. If some of the hermaphrodites were shifted at L4 (the time at which spermatogenesis occurs) the authors might have underestimated the effect of the mutation on male meiotic centriole duplication, as some of the spermatocytes might have executed duplication at the permissive temperature. All hermaphrodites should have been shifted to restrictive temperature at the L1 stage. Related to this, the authors should present the data as percent of cells with monopolar spindles at the first cell cycle (meiotic defect) and percent of cells with monopolar spindles at the second cell cycle (mitotic defect) so that the reader can easily determine the relative effects of the mutation. I bring this up because the table is a bit misleading in that it appears to show that only 18% of embryos show a duplication defect at the two-cell stage in *or452* hermaphrodites. However, looking closer, the 18 percent refers to the total number of embryos scored. If I understand the data correctly, the authors only scored 10 embryos at the two-cell stage and of these 6 showed at least one monopolar spindle. It would be best to score the first and second cell cycles separately with each expressed as a percentage of cells (not embryos) with monopolar spindles. Finally, there are no proper controls for the wild-type x mutant crosses (last two rows of table). The proper controls are wild-type males crossed to *fem-1(hc17*ts) females and *or452* males crossed to *or453; hc17* females. Related to this last point, why do the authors not discuss the apparent rescue of the centriole duplication defect at the two-cell stage by mating mutant females to wild-type males (8%)? Does this mean that the effect of the *or452* mutation is largely paternal?"

Increased n would be helpful, since the frequency at which the 2nd division phenotypes were observed (1/13 and 1/26), particularly in the important *or452* female x WT experiment was very low.

2) Include a description of the RNAi phenotype. This is important even if it does not recapitulate the mutant phenotypes. RNAi should be performed in the GFP::SAS-7 expressing worms so the effect of the RNAi on protein levels at centrioles can be ascertained in parallel to the phenotypes. In particular, it would be important to assess both centriole duplication and the establishment of polarity (perhaps using the flow assay in Figure 5) in the RNAi embryos. It would also be useful to know if RNAi targeting the mutant protein could rescue any of the *or452* mutant phenotypes since the *or452*ts allele is not null and the mutant protein may retain the ability to localize to centrioles.

3) Assess the fate of the mutant protein. This could be achieved in multiple ways – through introduction of a tagged mutant transgene followed by RNAi to deplete the endogenous protein, through CRISPR tagging of the mutant locus (if the background makes this difficult, it might be possible to introduce a re-encoded WT transgene and then edit the endogenous mutant locus), or by immunofluorescence if an antibody to SAS-7 is available. This experiment would be useful to determine whether the deleted C-terminal region only affects the interaction between SAS-7 and SPD-2 or also the ability of SAS-7 to target to centrioles and would make a major contribution towards clarifying multiple aspects of the data.

4) Determine whether deleting the region absent in the *or452* mutant disrupts SPD-2 binding in the context of the full-length protein in the 2-hybrid assay to rule out the possibility that other regions of SAS-7 can also mediate an interaction with SPD-2.

5) Quantification of the number of embryos examined and the% exhibiting different outcomes needs to be provided for all imaging-based experiments. All of the image panels also need scale bars.

Encouraged but not essential:

1) Address whether the *sas-7* mutant affects centriole assembly versus stability. The *sas-7* mutant phenotype is superficially similar to that of SAS-1/C2CD3, which is thought to function in centriole stability (von Tobel et al., PLOS Genet 2014). In both cases, there is a much stronger paternal than maternal requirement, and monopolar spindles occur much less frequently than in *spd-2, zyg-1* or *sas-5* mutants. From the data presented it is not entirely clear if new centrioles do not form or are unstable. This could be addressed by monitoring the fate of GFP:SAS-4 or SAS-6-labeled sperm centrioles in a *sas-7* mutant background.

---

## [Author Response]

Essential revisions:

1) Whether or not SAS-7 makes a significant contribution to centriole duplication, and in what contexts (male germline vs. mitotic embryos) needs to be clarified. The prominent phenotype in the or452ts mutant is the presence of monopolar first embryonic divisions, which results from the production of sperm that have only one centriole. This phenotype could arise from a defect in centriole duplication in the male germline, or from another problem related to loss of the paddlewheel that limits the ability of centrioles to be packaged into sperm. More work monitoring centriole duplication and fate in the male germline would be needed to sort this out if the authors want to retain the focus of the paper on a role for SAS-7 in centriole duplication. Whether SAS-7 controls centriole duplication during the mitotic divisions in the embryo is a separate question from its role in the male germline. From the data presented, the role in centriole duplication during mitotic divisions appears to be quite minimal. If the role of SAS-7 in centriole duplication in embryos is relatively minor, the manuscript (and title) should be rewritten to focus the roles of SAS-7 in assembly of the paddle wheel, in controlling SPD-2 recruitment, and in controlling the ability of the centrosome to communicate centrosome position to the cortex to establish embryonic polarity.

To analyze whether centriole duplication in sperm meiosis is defective, we performed the following experiments. First, we observed GFP-SAS-7 at centrioles throughout male meiosis I and II (Figure 2—figure supplement 1). Second, we counted the number of spindle poles in sperm meiosis I in *sas-7(or452*ts) mutants (Figure 2—figure supplement 1). Third, we stained with anti-SAS-4 to count the number of centrioles after sperm meiosis II and after the first embryonic mitosis in *sas-7(or452*ts) mutants (Figure 2). We found that in *sas-7(or452*ts) mutants centriole duplication during sperm meiosis I was completely normal, while that during sperm meiosis II and during thefirst embryonic mitosis were defective, resulting in monopolar spindles during the first embryonic mitosis and later mitoses, respectively. As the *or452*ts mutation is a reduction of function mutation, and the stronger allele *or1945* results in adult sterility, we do not know whether sperm meiosis I requires SAS-7 or not. Similarly, strong alleles of other centriole duplication genes also result in adult sterility. Therefore, it is possible that centriole duplication in the mitotic germline and during sperm meiosis I require SAS-7 but are not defective in *or452*ts due to residual SAS-7 function. We have revised the text to describe this conclusion as follows:

“While our results indicate that sperm meiosis II is defective, we did not detect any centriolar duplication defects during sperm meiosis I in *sas-7(or452*ts) mutants (Figure 2—figure supplement 1). […] Taken together, our results suggest that the reduced SAS-7 activity resulted in the severe centriole duplication defects in both sperm meiosis II and embryonic mitosis.”

Furthermore, in the original Table 1, we incorrectly reported the total number of embryos scored for monopolar spindles in early embryos, which resulted in under-reporting the penetrance of the mitosis defects. Now we have separately counted the number of monopolar spindles during the first mitosis, and during the second and third mitoses. In addition, we have now counted the number of monopolar spindle inthe fourth to seventh mitoses to further document the requirements for maternal SAS-7 during mitosis. As shown in the revised Table 1, *sas-7(or452*ts) mutants are highly penetrant for monopolar spindle defects both during the first embryonic mitosis (due to defects during sperm meiosis II) and during subsequent mitoses (due to maternal requirements for SAS-7).

Clarification would also include addressing the points raised in the following comment from reviewer #2:

"To clarify the extent to which the or452 mutant affects maternal and paternal duplication it is important to clearly indicate at what developmental stage the or452 hermaphrodites are shifted to the restrictive temperature. Table 1, footnote b specifies L1-L4. If some of the hermaphrodites were shifted at L4 (the time at which spermatogenesis occurs) the authors might have underestimated the effect of the mutation on male meiotic centriole duplication, as some of the spermatocytes might have executed duplication at the permissive temperature. All hermaphrodites should have been shifted to restrictive temperature at the L1 stage. Related to this, the authors should present the data as percent of cells with monopolar spindles at the first cell cycle (meiotic defect) and percent of cells with monopolar spindles at the second cell cycle (mitotic defect) so that the reader can easily determine the relative effects of the mutation. I bring this up because the table is a bit misleading in that it appears to show that only 18% of embryos show a duplication defect at the two-cell stage in or452 hermaphrodites. However, looking closer, the 18 percent refers to the total number of embryos scored. If I understand the data correctly, the authors only scored 10 embryos at the two-cell stage and of these 6 showed at least one monopolar spindle. It would be best to score the first and second cell cycles separately with each expressed as a percentage of cells (not embryos) with monopolar spindles. Finally, there are no proper controls for the wild-type x mutant crosses (last two rows of table). The proper controls are wild-type males crossed to fem-1(hc17ts) females and or452 males crossed to or453; hc17 females. Related to this last point, why do the authors not discuss the apparent rescue of the centriole duplication defect at the two-cell stage by mating mutant females to wild-type males (8%)? Does this mean that the effect of the or452 mutation is largely paternal?"

Increased n would be helpful, since the frequency at which the 2nd division phenotypes were observed (1/13 and 1/26), particularly in the important or452 female x WT experiment was very low.

We apologize for our incorrect reporting of the data in the original Table 1 and have fixed this problem in the revised manuscript. First, we re-performed the all temperature upshift experiments from L1 stage animals as suggested. As a result, now *or452*ts mutants shows 71% sperm meiosis II defects (monopolar spindles in P_0_) and 57% first mitosis defects (monopolar spindles in P_1_ or AB). Furthermore, 100% of the embryos that did not have monopolar spindles in P_0_, P_1_ or AB did show monopolar spindles in later divisions. Also, we performed Male x Female crosses to show the paternal and maternal contributions of *sas-7* gene and now include the control experiment (WT male x *fem-1* female). As reviewer 2 pointed out, after female *sas-7(or452*ts) mutants and wild-type males were crossed, defects in sperm meiosis II (monopolar spindle in P_0_ cell) were completely rescued, and 2-cell stage embryos only showed 4% defects. However, 92% of the fourth to seventh mitosis showed defects, suggesting that paternal SAS-7 can rescue sperm meiosis and the first mitosis but not later stage mitosis. On the other hand, when we crossed wild-type female and male *sas-7(or452*ts) mutants, 65% of sperm meiosis II events were defective (monopolar spindle in P_0_ cell), and later mitotic divisions were completely normal. Based on these data, we concluded that paternal SAS-7 mainly contributes to sperm meiosis andthe first mitosis and maternal SAS-7 mainly contributes to the first mitosis and later mitotic divisions.

We thank the reviewers for pointing out that we needed to clarify our understanding of the requirements for both paternal and maternal SAS-7; these changes significantly improve our manuscript and provide some of the best insight into these issues for any centriolar duplication gene, given that most studies of other genes have been based on RNAi knockdowns that likely only affect maternal protein expression.

Finally, we want to point out that we present these data by combining numbers for the second and third divisions, and for the fourth-seventh divisions, reported as defects per embryo, and not for each individual cell. For example, defects in the fourth-seventh mitoses indicate a defect in at least one of the cells in a single embryo. While the reviewer asked for number of defects per cell, we think this approach satisfies the reviewer’s intent, by reporting defects in each set of cell cycles separately (first division; second + third division; fourth – seventh division), and no longer combine all cell cycles from each embryo as in the original submission.

2) Include a description of the RNAi phenotype. This is important even if it does not recapitulate the mutant phenotypes. RNAi should be performed in the GFP::SAS-7 expressing worms so the effect of the RNAi on protein levels at centrioles can be ascertained in parallel to the phenotypes. In particular, it would be important to assess both centriole duplication and the establishment of polarity (perhaps using the flow assay in Figure 5) in the RNAi embryos. It would also be useful to know if RNAi targeting the mutant protein could rescue any of the or452 mutant phenotypes since the or452ts allele is not null and the mutant protein may retain the ability to localize to centrioles.

We now include RNAi knockdown data. At the time of our original submission, we knew that RNAi against SAS-7 was not effective, and therefore we made additional mutants using CRISPR/Cas9 that confirmed requirements for SAS-7 during centriole duplication as originally reported. We now also show in Figure 1—figure supplement 2 that *sas-7* RNAi neither eliminates the GFP::SAS-7 signal nor results in highly penetrant embryonic lethality. Nevertheless, by using GFP RNAi and looking at the F2 embryos produced by escaper fertile F1s, we observed monopolar spindles in early embryos and the loss of cell polarity (equal first division phenotype) in one embryo. However, because most escaper F1s after GFP RNAi were adult sterile, we could analyze only a limited number of embryos.

With respect to the concern that *sas-7(or452ts)* might have a dominant effect, we note the following points: (i) *sas-7(or452ts)/nDf40* (large deletion lacking entire sas-7 gene) did not result in any phenotype rescue but rather resembled embryos from homozygous *sas-7(or452ts)* (Table 1); (ii) a wild-type *sas-7* gene could rescue *sas-7(or452ts)* lethality (Figure 1); (iii) substantially reduced levels of cytoplasmic and centriolar GFP-SAS-7(or452ts) were observed compared to wild-type GFP-SAS-7 (this third point being new data). These results are all consistent *or452*ts being a reduction of function mutation. However, as originally reported *or452*ts is not null as the early frameshift mutation *or1945* results in adult sterility.

3) Assess the fate of the mutant protein. This could be achieved in multiple ways – through introduction of a tagged mutant transgene followed by RNAi to deplete the endogenous protein, through CRISPR tagging of the mutant locus (if the background makes this difficult, it might be possible to introduce a re-encoded WT transgene and then edit the endogenous mutant locus), or by immunofluorescence if an antibody to SAS-7 is available. This experiment would be useful to determine whether the deleted C-terminal region only affects the interaction between SAS-7 and SPD-2 or also the ability of SAS-7 to target to centrioles and would make a major contribution towards clarifying multiple aspects of the data.

By CRISPR/Cas9 we made the GFP-fusion of *sas-7(or452ts)* (Figure 4). GFP-SAS-7(or452ts) localizes to centrioles, but at a greatly reduced level. As cytoplasmic SAS-7 level also is reduced, we conclude that the expression level of SAS-7 was reduced by the mutation.

4) Determine whether deleting the region absent in the or452 mutant disrupts SPD-2 binding in the context of the full-length protein in the 2-hybrid assay to rule out the possibility that other regions of SAS-7 can also mediate an interaction with SPD-2.

We performed yeast-two hybrid assay to address this point (Figure 5). We found that SAS-7(*or452*ts) full length construct retains SPD-2 interacting ability, and we identified an additional SPD-2 binding domain in the C-terminus of the SAS-7 protein. These data, along with reduced expression level of GFP-SAS-7(or452ts), are consistent with the *sas-7(or452ts)* mutation causing a substantial reduction in function while still retaining some activity that allows for germline proliferation (with the stronger allele *or1945* resulting in adult sterility and precluding analysis of a stronger loss of function in the early embryo). We think that these new data lend further support to our model for SAS-7 function as illustrated in Figure 7.

5) Quantification of the number of embryos examined and the% exhibiting different outcomes needs to be provided for all imaging-based experiments. All of the image panels also need scale bars.

We have added this information to all figures.

Encouraged but not essential:

1) Address whether the sas-7 mutant affects centriole assembly versus stability. The sas-7 mutant phenotype is superficially similar to that of SAS-1/C2CD3, which is thought to function in centriole stability (von Tobel et al., PLOS Genet 2014). In both cases, there is a much stronger paternal than maternal requirement, and monopolar spindles occur much less frequently than in spd-2, zyg-1 or sas-5 mutants. From the data presented it is not entirely clear if new centrioles do not form or are unstable. This could be addressed by monitoring the fate of GFP:SAS-4 or SAS-6-labeled sperm centrioles in a sas-7 mutant background.

In the new Table 1, we show that *sas-7(or452ts)* mutants have defects in both sperm meiosis and during embryonic mitosis, and that both paternal and maternal contributions are essential. Please also see response #1 above for a more detailed summary of these results. In *sas-1* mutants, centriole microtubules are almost absent, which is not true for *sas-7(or452*ts) mutants based on our EM data as shown in Figure 3. Given this substantial difference in mutant centriole ultrastructures, we do not think SAS-7 is required for the centriole stability.